# Epigenomic diagnosis and prognosis of Acute Myeloid Leukemia

Francisco Marchi [1], Vivek M. Shastri [1], Richard J. Marrero [1], Nam H. K. Nguyen [1], Antonella Öttl[1], Ann-Kathrin Schade[1], Marieke Landwehr[1], Olga Krali [2,3], Jessica Nordlund [2,3], Matin Ghavami[4], Fernando Sckaff[1], Vikash K. Mansinghka[5], Xueyuan Cao [6], William Slayton[7,8], Petr Starostik[8,9], Christopher R. Cogle[8,10], Raul C. Ribeiro [11], Jeffrey E. Rubnitz[11], Jeffery Klco [12], Abdelrahman Elsayed[13], Alan S. Gamis[14], Timothy J. Triche Jr. [15], Rhonda Ries [16], E. Anders Kolb[17], Richard Aplenc[18,19], Todd Alonzo[20], Stanley Pounds [13], Soheil Meshinchi[16] & Jatinder K. Lamba [1,8] ✉

Despite the critical role of DNA methylation, clinical implementations harnessing its promise have not been described in acute myeloid leukemia. Utilizing DNA methylation from 3314 leukemia patient samples across 11 harmonized cohorts, we describe the Acute Leukemia Methylome Atlas, which includes robust models capable of accurately predicting AML subtypes. A genome-wide prognostic model as well as a targeted panel of 38 CpGs significantly predict five-year survival in our pediatric and adult test cohorts. To accelerate rapid clinical utility, we develop a specimen-to-result protocol that uses long-read nanopore sequencing and machine learning to characterize patients' whole genomes and epigenomes. Clinical validation on patient samples confirms high concordance between epigenomic signatures and genomic lesions, though uniquely rare karyotypes remained challenging due to limited available training data. These results unveil the potential for increased affordability, speed, and accuracy for patients in need of complex molecular diagnosis and prognosis.

Acute Myeloid Leukemia (AML) is a devastating disease associated with high morbidity and mortality. Decades of fundamental research demonstrate that leukemic stem cells are highly heterogeneous, elusive, and ever evolving, so translating that many layers of complexity into clinical care has been a monumental challenge[1]. Compromised epigenomic integrity hallmarks the disease in the young and the old, which warrants a comprehensive rather than isolated study. Diagnostic subtyping and risk stratification are foundational pillars of treatment

[1]Pharmacotherapy and Translational Research, College of Pharmacy, University of Florida, Gainesville, FL, USA. [2]Department of Medical Sciences, Molecular Precision Medicine, Uppsala University, Uppsala, Sweden. [3]Science for Life Laboratory, Uppsala University, Uppsala, Sweden. [4]Department of Electrical Engineering and Computer Science, MIT, Cambridge, MA, USA. [5]Department of Brain and Cognitive Sciences, MIT, Cambridge, MA, USA. [6]Department of Health Promotion and Disease Prevention, University of Tennessee Health Science Center, Memphis, TN, USA. [7]Department of Pediatrics Hematology, University of Florida, Gainesville, FL, USA. [8]University of Florida Health Cancer Center, Gainesville, FL, USA. [9]Department of Pathology, University of Florida Cancer Center, Gainesville, FL, USA. [10]Department of Medicine, College of Medicine, University of Florida, Gainesville, FL, USA. [11]Department of Oncology, St. Jude Children's Research Hospital, Memphis, TN, USA. [12]Department of Pathology, St. Jude Children's Research Hospital, Memphis, TN, USA. [13]Department of Biostatistics, St. Jude Children's Research Hospital, Memphis, TN, USA. [14]Hematology/Oncology/BMT, Children's Mercy Kansas City, Kansas City, MO, USA. [15]Department of Epigenetics, Van Andel Institute, Grand Rapids, MI, USA. [16]Translational Science and Therapeutics, Fred Hutchinson Cancer Center, Seattle, WA, USA. [17]Leukemia & Lymphoma Society, Washington D.C., USA. [18]Department of Pediatrics, Perelman School of Medicine at the University of Pennsylvania, Philadelphia, PA, USA. [19]Division of Oncology, Children's Hospital of Philadelphia, Philadelphia, PA, USA. [20]University of Southern California Keck School of Med, Los Angeles, CA, USA. ✉e-mail: jatinderklamba@ufl.edu

regimens, clinical trial design, and systematically informed therapy decisions. However, despite extensive evidence pointing to the central role of DNA methylation in marking proliferation and differentiation of AML[2–4], current guidelines rely on a selective list of genomic lesions for diagnosis and prognosis, which does not capture disease heterogeneity and pleiotropy. Due to a lack of feasible technological solutions, implementation of epigenomics in clinical hematology/oncology settings are lacking, and so are studies providing evidence to its promise.

Genomic and epigenomic analyses are plagued by the overwhelming amount of data they generate, outputting an odyssey of variants of unknown significance, obfuscating clinically relevant information. To provide refined and impactful data with meaningful interpretation, novel machine learning algorithms have shown success in detecting patterns invisible to human curation. Seminal work in recent years demonstrated the use of dimension reduction techniques to define diagnoses of central nervous system tumors and sarcomas[5,6], which led to the WHO recommending genomic classification systems over morphology, despite full awareness that most clinicians only currently have access to the latter[7]. To address that, long-read DNA sequencing technologies now allow for the detection of cytosine modifications natively, which enables concomitant genome and epigenome sequencing[8,9].

In this work, we present a robust framework that combines large-scale epigenomic data harmonization, machine learning, web-interface, and innovative sequencing technologies to advance the diagnostic and prognostic landscape of AML. Our findings support the integration of epigenomic profiling into the hematology/oncology unit setting, offering a pathway toward more personalized and effective trial designs and treatment modalities for patients with AML.

## Results

### The acute leukemia methylome atlas maps hematopoietic heterogeneity

The acute leukemia methylome atlas (ALMA) was built using an unsupervised dimension reduction algorithm named Pairwise Controlled Manifold Approximation (PaCMAP), which compressed methylation values of 331,556 CpGs into two dimensions for visualization and five dimensions for subtype and risk classification (Fig. 1a). Samples with similar epigenomes clustered together, globally representing the following cell populations of origin: AML ($n = 1221$), ALL ($n = 700$), MDS-related or secondary myeloid neoplasms (MDS-like) ($n = 223$), acute promyelocytic leukemia (APL) ($n = 31$), mixed phenotype leukemia (MPAL) ($n = 48$), and otherwise-normal control ($n = 251$). These were further subdivided into local structures that overlapped with the heterogeneous clinical subtypes within the disease domains described above (Fig. 1b and Supplementary Table 1).

To facilitate data navigation and interpretation, we developed an interactive visualization tool for ALMA and provided it in the form of a web-app: https://f-marchi.github.io/UF-LambaLab-ALMA-app/. This tool enables users to zoom, select, drag, and save specific areas of ALMA, as well as change tabs to compare the data under different clinical annotations. They can also select a particular cluster and observe its predicted risk profile. As a case study, we describe the *AML with t(v;11q23); KMT2A-r* cluster with samples colored according to different clinical annotations (Supplementary Fig. 1). In addition, we incorporated the recently reported expanded risk group for the COG-AAML1831 trial[10]. ALMA enabled the creation of two supervised models: *ALMA Subtype* and *AML Epigenomic Risk*. The performance of the models is described below, and the patient characteristics distribution is described in Supplementary Table 2.

### ALMA Subtype accurately stratifies WHO2022 subtypes by DNA methylation

The diagnostic model within *ALMA* named *ALMA Subtype* is a LightGBM supervised multi-class classifier that uses five PaCMAP coordinates to predict 27 WHO 2022 subtypes plus otherwise-normal control (Fig. 2a). Applying these subtypes to the discovery dataset with available WHO 2022 diagnosis annotation ($n = 2471$) generated a per-class 5-fold cross-validation (CV) concordance score ranging from 0.74 for *AML with t(v;11q23); KMT2A-r* and 0.8 for *MDS-related, secondary myeloid*, both highly heterogeneous, ambiguous subtypes, to a perfect 1.00 in sixteen other categories. The combined annotated test cohorts AML02, AML08, and NOPHO AML ($n = 180$) showed per class concordance scores of 0.91 for AML with *inv(16); t(16;16); CBFB::MYH11* ($n = 35$), 0.70 for *AML with mutated CEBPA* ($n = 10$), 1.00 for *AML with mutated NPM1* ($n = 3$), 1.00 for *AML with t(6;9;DEK::NUP214* ($n = 1$), 1.00 for *AML with t(8;16); KAT6A::CREBBP* ($n = 2$), 0.98 for AML with *t(8;21); RUNX1::RUNX1T1* ($n = 48$), 0.83 for *AML with t(v;11); KMT2A-r* ($n = 72$), 1.00 for *APL with t(15;17); PML::RARA* ($n = 4$), and 0.60 for *MDS-related, secondary myeloid* ($n = 5$) (Fig. 2b). Sankey plots in Fig. 3 shows comparison of the WHO2022 diagnosis and ALMA subtype in the discovery (Fig. 3a) and the cohorts (Fig. 3b, c), additionally ALMA subtype in patients with no WHO2022 classification are shown in Fig. 3d–f.

Perhaps unintuitively but powerfully, inclusion of ALL, APL, MDS, and control samples enhanced the overall capacity of the classifier to predict specific AML subtypes. Notably, the resulting classifier allowed prediction of clinical diagnosis of 840 samples in the discovery cohort and 96 samples in the validation cohort for which the WHO 2022 diagnosis annotation was unavailable. These samples contained inconclusive descriptions such as *Normal Karyotype*, *Other*, *Complex Karyotype*, or relied on unclassified fusions or morphological descriptions with ambiguous equivalent WHO 2022 diagnoses, such as in FAB = M0, M1, M6, and M7 (Supplementary Fig. 2).

Overall metrics for the discovery/training set (5-fold CV, $n = 2471$) included an accuracy of 0.896, a weighted F1 score of 0.927, and a Cohen's Kappa of 0.888. In the test sets, the AML02,08 cohort ($n = 104$) achieved an accuracy of 0.875, a weighted F1 score of 0.924, and Cohen's Kappa of 0.825, while the NOPHO AML cohort ($n = 76$) showed an accuracy of 0.895, a weighted F1 score of 0.925, and Cohen's Kappa of 0.869. These results suggest negligible overfitting and rigorous performance in independently processed test datasets.

### AML Epigenomic Risk accurately predicts 5-year overall survival in AML

Similar to its diagnostic counterpart, the prognostic model named *AML Epigenomic Risk* is a LightGBM supervised classifier developed using the 331556 CpGs compressed into five PaCMAP coordinates from ALMA to predict the probability of death within 5 years for AML patients. In the discovery cohort ($n = 946$), *AML Epigenomic Risk*[high] demonstrated a markedly poorer OS (HR = 4.40; 95% CI = 3.45, 5.61; $P < 0.0001$) and EFS (HR = 2.39; 95% CI = 2.00, 2.86; $P < 0.0001$) in comparison to *AML Epigenomic Risk*[low] (Fig. 4a). In the AML02,08 test cohort ($n = 200$), patients within *AML Epigenomic Risk*[high] group had poor OS (HR = 4.20 95% CI = 2.36, 7.45; $P < 0.0001$) and EFS (HR = 3.26; 95% CI = 2.07, 5.15; $P < 0.0001$) as compared to the *AML Epigenomic Risk*[low] group (Fig. 4b). Distribution of patient characteristics by *AML Epigenomic Risk* groups in discovery and AML02,08 test cohort is summarized in Supplementary Table 3. In NOPHO pediatric AML test cohort, AML Epigenomic Risk[high] was significantly associated with poorer 5-year OS (HR = 2.48; 95% CI = 1.34, 4.61; $P = 0.0039$, Supplementary Fig. 3a). To elucidate, however, whether adult AML patients could also benefit from AML Epigenomic Risk, we evaluated the model in TCGA and Beat AML cohorts and consistent with pediatric cohorts, OS in *AML Epigenomic Risk*[high] group was significantly worse in both TCGA (HR = 1.72; 95% CI = 1.21, 2.43; $P = 0.0022$) and Beat AML (HR = 2.02; 95% CI = 1.30, 3.15; $P = 0.0017$) cohorts (Supplementary Fig. 3b, c).

MRD1 was available in discovery and AML02,08 test cohorts, and in both cohort,s *AML Epigenomic Risk* was significantly associated with induction 1 MRD (discovery cohort, $P < 0.001$; AML02,08 test cohort,

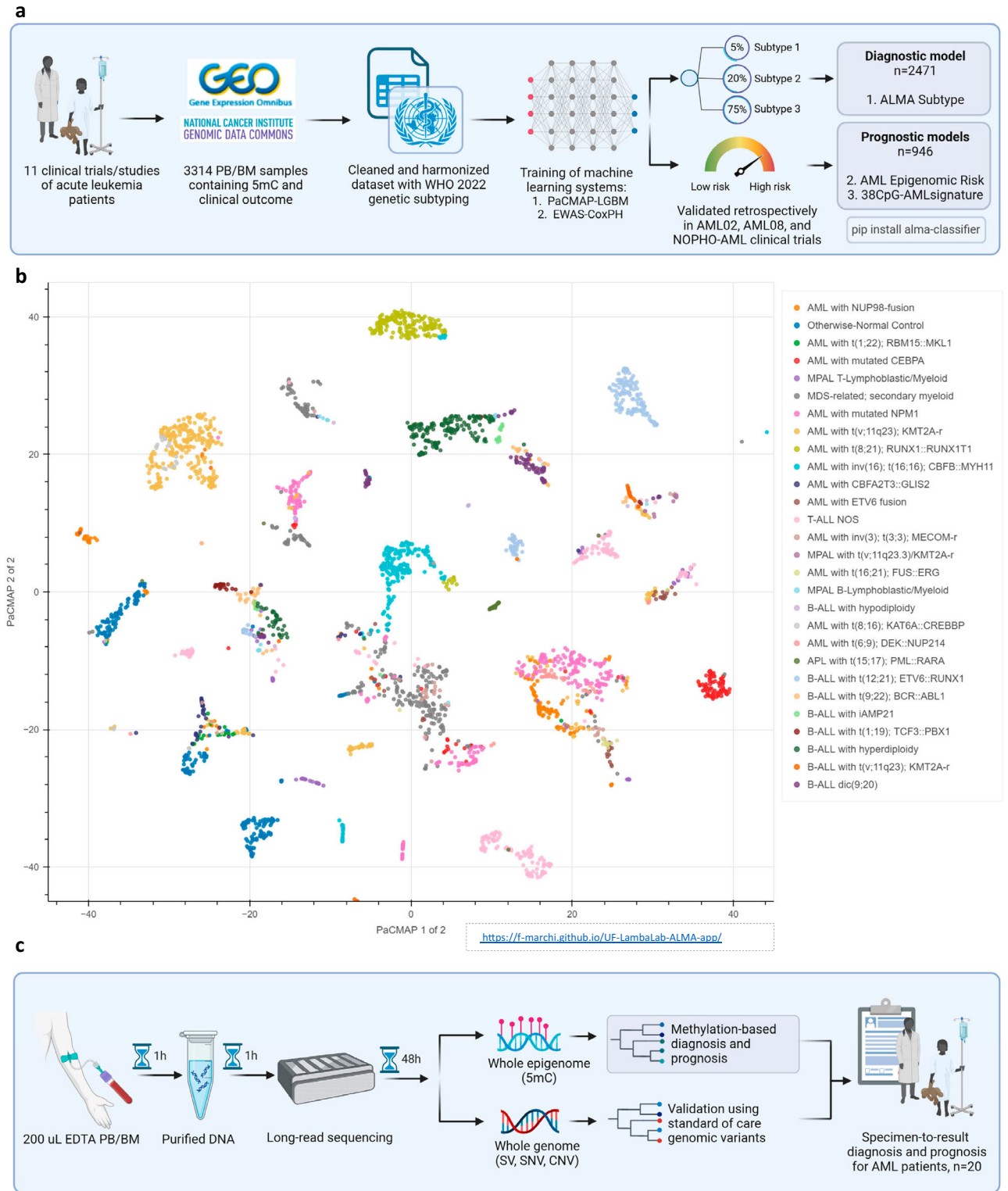

**Fig. 1 | The Acute Leukemia Methylome Atlas Study. a** Overall design of the computational pipeline. **b** Acute Leukemia Methylome Atlas (ALMA) defined by a dimension reduction algorithm, PaCMAP, in the overall population. Each data point is a patient sample deriving from bone marrow or peripheral at diagnosis, relapse, or remission. As an unsupervised model, only 5mC values of 331,556 CpGs were used to define the clusters. **c** Clinical implementation of a long-read nanopore sequencing pipeline. Discovery (training) raw data analyzed in this study were obtained from Gene Expression Omnibus (GEO) under accession codes GSE190931, GSE124413, GSE133986, GSE159907, GSE152710, GSE49031, GSE147667, as well as

from Genomic Data Commons (GDC) under categories GDC-TARGET-AML, GDC-TCGA-AML, GDC-TARGET-ALL. AML acute myeloid leukemia, ALL acute lymphoblastic leukemia, MDS myelodysplastic syndrome, MPAL mixed phenotype acute leukemia, APL acute promyelocytic leukemia, PaCMAP pairwise controlled manifold approximation, WHO World Health Organization, PB peripheral blood, BM bone marrow, EDTA Ethylenediaminetetraacetic acid, HMW gDNA high molecular weight genomic DNA. Panels (**a** and **c**) were created in BioRender. Marchi, F. (2025) https://BioRender.com/786kvbb.

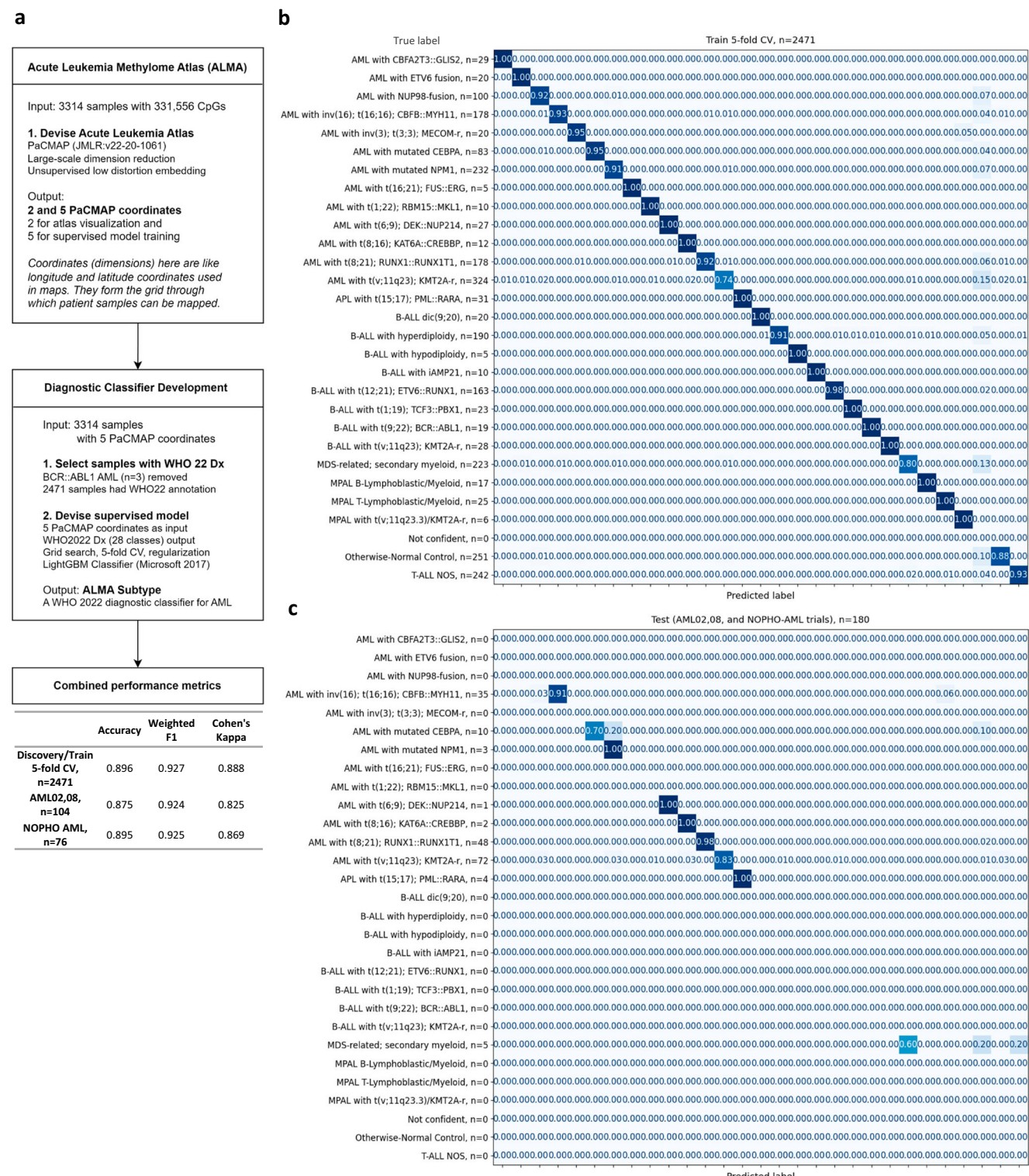

**Fig. 2 | Development and benchmarking of ALMA Subtype. a** Diagram of the study design for ALMA and ALMA Subtype. **b** Sample count by WHO 2022 Diagnosis subtypes in train (*n* = 2471) and (**c**) test (AML02, AML08, and NOPHO AML; *n* = 180). Multi-class confusion matrices with normalized values for each possible prediction, along with the number of samples available, are displayed. Not confident: < 50% confidence for a subtype prediction.

*P* = 0.033). *AML Epigenomic Risk* groups further significantly predicted EFS and OS within MRD1-positive and MRD1-negative subgroups in both discovery and test cohorts (Fig. 5a, b).

*AML Epigenomic Risk* remained as an independent predictor of OS (HR = 3.86; 95% CI = 2.74, 5.44; *P* < 0.0001) (Fig. 4c) and EFS (HR = 1.78; 95% CI = 1.40, 2.27; *P* = 0.0009) (Fig. 4d) in multivariable analysis after

adjusting for MRD1 status, risk group, FLT3 status, leucocyte counts at diagnosis, BM blast % at diagnosis, and age groups in discovery cohort. Similarly, in the AML02,08 test cohort, *AML Epigenomic Risk* remained an independent predictor of OS (HR = 2.83; 95% CI = 1.42, 5.64; *P* = 0.0032) (Fig. 4e) and EFS (HR = 2.65; 95% CI = 1.49, 4.71; *P* = 0.0009) (Fig. 4f).

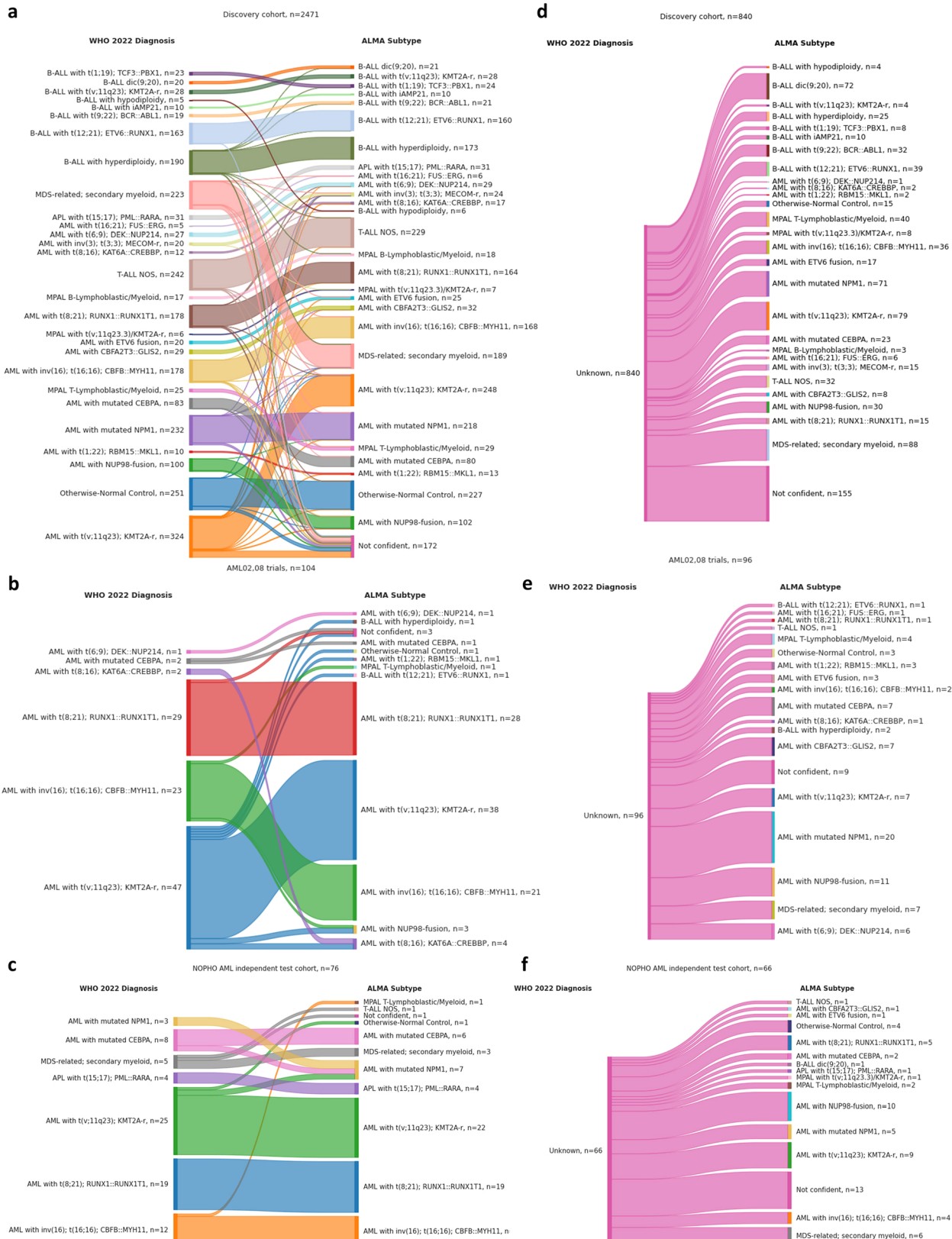

**Fig. 3 | ALMA Subtype classification according to WHO 2022 subtypes.** Sankey diagrams displaying discovery cohort comparison between WHO 2022 diagnosis and ALMA Subtype for samples with (**a**) or without (**d**) WHO 2022 clinical annotation available. The width of the bands indicates the number of patient samples in each category. Same comparison applied to test cohorts (**b**, **e**) AML02,08 ($n = 200$) and (**c**, **f**) NOPHO AML ($n = 142$), describing samples with or without the WHO 2022 annotation available. Individual n numbers are indicated in the figures.

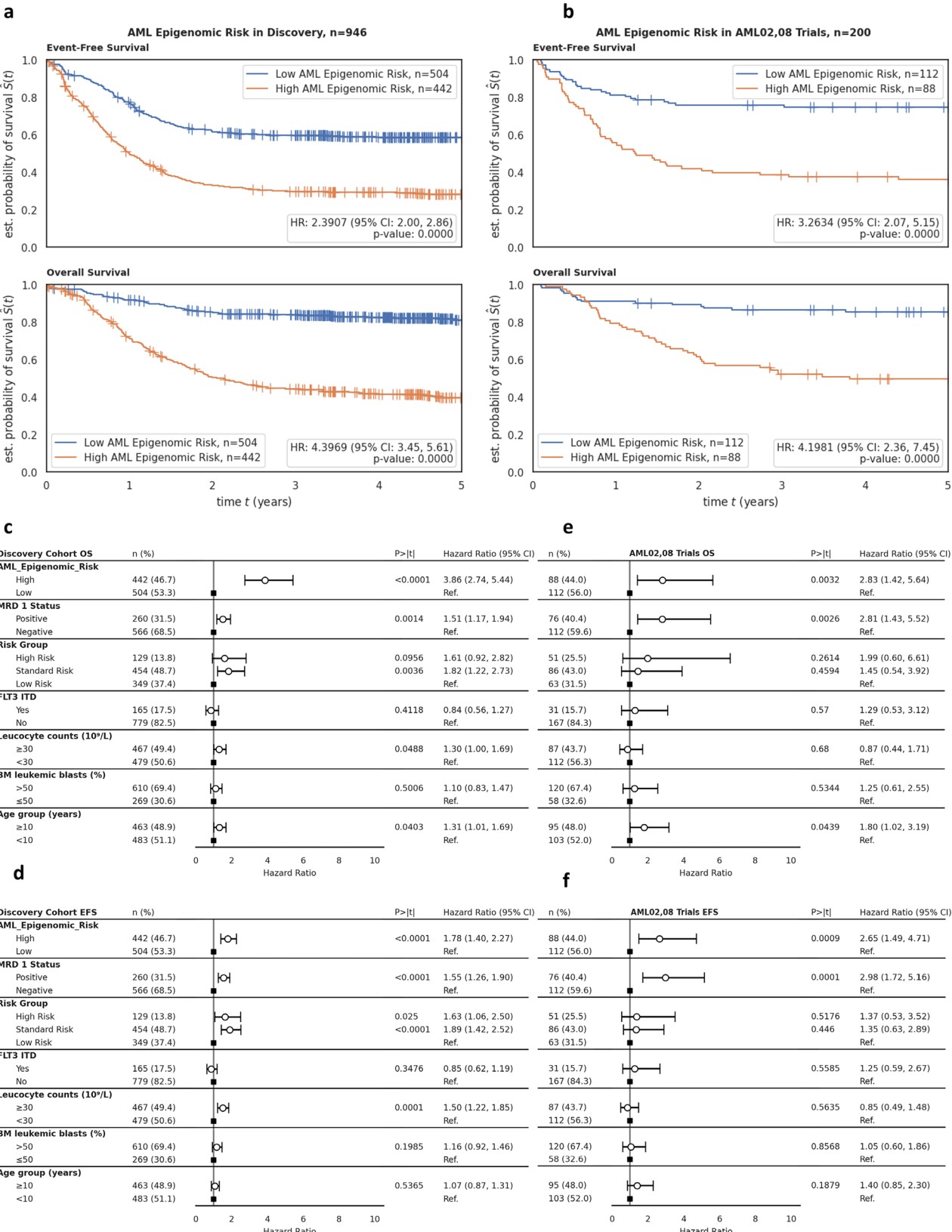

**Fig. 4 | Patient outcomes and multivariate analyses of AML Epigenomic Risk groups.** Patient outcomes by *AML Epigenomic Risk* groups with in MRD1 subgroups in (**a**) the discovery cohort and (**b**) the AML02,08 test cohort with EFS (top) and OS (bottom) of *AML Epigenomic Risk*high (orange) and *AML Epigenomic Risk*low (blue). Multivariate analysis adjusting for other confounding variables of OS in (**c**) discovery cohort (HR = 3.86; *P* < 0.0001) and (**e**) AML02,08 test cohort (HR = 2.83;

*P* = 0.0032). Analysis for EFS was also performed in (**d**) the discovery cohort (HR = 1.78; *P* < 0.0001) and (**f**) the AML02,08 test cohort (HR = 2.65; *P* = 0.0009). Hazard ratios derive from Cox PH regression with two-sided hypothesis tests. Individual n numbers are indicated in the figures. MRD 1 minimal residual disease at end of first induction, FLT3 ITD FMS-like tyrosine kinase-3 internal tandem duplication, CI confidence interval, Ref. reference.

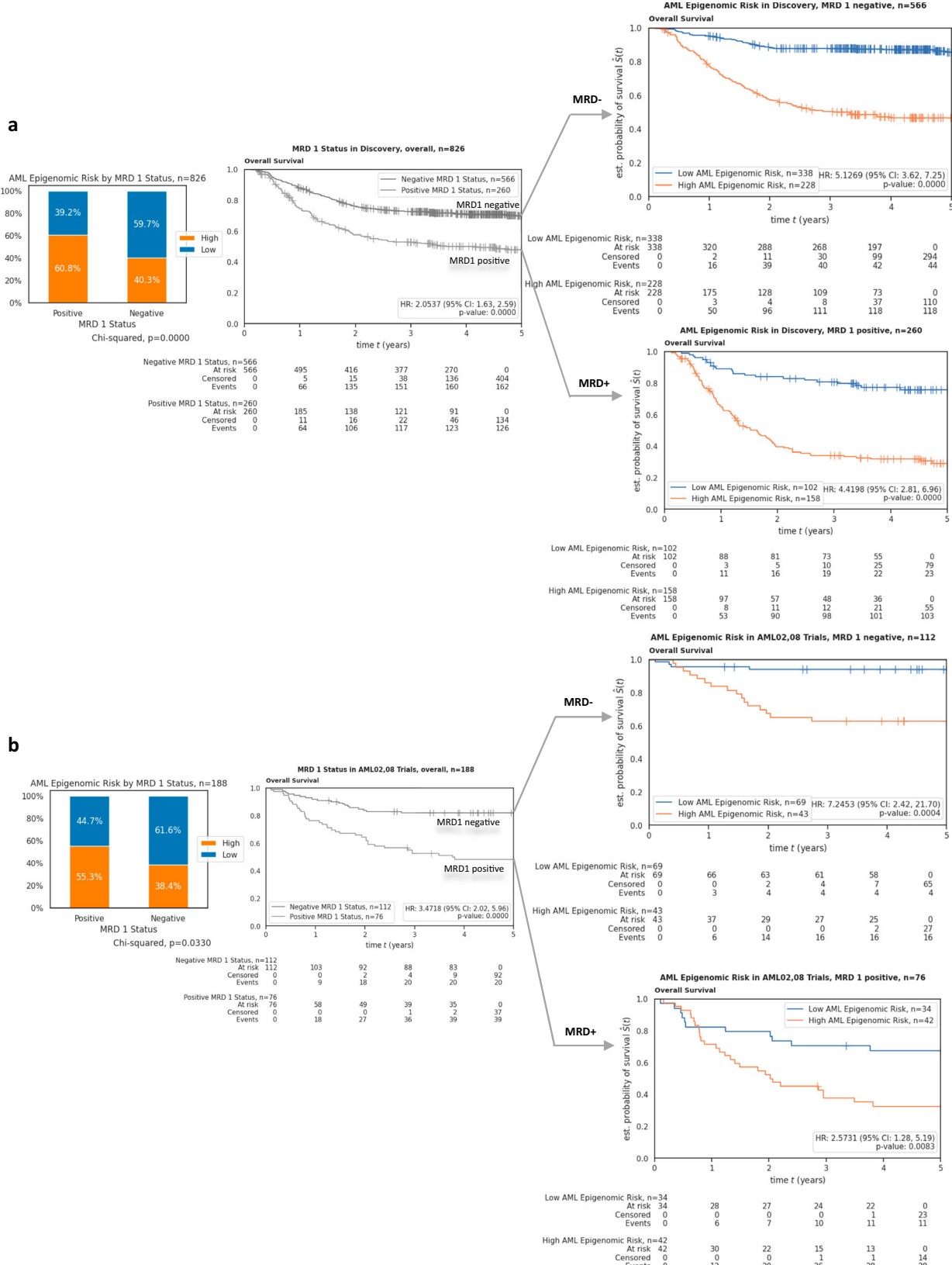

**Fig. 5 | Patient outcomes by Epigenomic Risk groups. within MRD1 positive negative groups.** Box plots show abundance of MRD1 positive and negative patients within *AML Epigenomic Risk* groups (chi-squared test; *p* < 0.0001 and *p* = 0.0330 for discovery and test, respectively. OS by MRD1 positive and MRD1 negative groups and OS within MRD1 positive and within MRD1 negative groups by *AML Epigenomic Risk groups* in (**a**) the discovery cohort and (**b**) AML02,08 test cohort. MRD 1, minimal residual disease at the end of the first induction. Only patients with MRD assessments were included in this analysis, which excludes patients who had events prior to the end of induction 1. Hazard ratios derive from Cox PH regression with two-sided hypothesis tests. Individual n numbers are indicated in the figures.

Given standard risk group patients show significant variability in outcome despite a lack of low or high-risk group features, we further evaluated *AML Epigenomic Risk* performance within the standard risk group only. Within 229 standard risk group patients at study entry from the discovery cohort, patients with *AML Epigenomic Risk*[high] had significantly poorer OS and EFS as compared to those with *AML Epigenomic Risk*[low] (OS HR = 4.37; 95% CI = 2.42, 7.89; $P < 0.0001$; EFS HR = 2.00; 95% CI = 1.36, 2.93; $P = 0.0004$). Similar results were observed in the 86 standard risk patients from the AML02,08 cohort (OS HR = 4.95, 95% CI = 1.88, 13.05; $P = 0.0012$; EFS HR = 2.08; 95% CI = 1.07, 4.04; $P = 0.0304$). Kaplan-Meier survival plots for EFS and OS by low, standard, and high-risk group categories are shown in Supplementary Fig. 4a–c.

## A 38-CpG prognostic signature predicts 5-year overall survival in AML

As an alternative approach, we describe a time-to-event-based signature of prognostic relevance that uses DNA methylation values of 38 CpGs, rather than 331556. Starting from the same prognostic cohort aforementioned ($n = 946$), we performed a Cox-PH-regression-based EWAS (Fig. 6a). EWAS adjusted for risk group assignment identified 200 CpGs significantly associated with 5-year time-to-death at a p-value threshold of 10e-5 (Fig. 6b). We chose OS as our primary outcome measure since EFS definitions are not consistent across the various trials described here.

The 200 CpGs identified in the EWAS were evaluated using a penalized Cox-PH model with 1000 iterations of 10-fold CV. Thirty-eight out of 200 CpGs consistently had non-zero coefficients in at least 95% of penalized fitting iterations (Fig. 6c). The *38-CpG AML signature* was thus defined by the linear combination of parameters calculated from the CpG M-values applied to the discovery cohort and test cohorts. Details on the coefficients, CpGs from EWAS, their corresponding genomic loci, and overlapping genes are described in Supplementary Table 4. To devise a binary score, median cutoff of − 2.043 was used, which allowed patients to be further categorized into high and low risk groups (Fig. 6d). In the discovery cohort, patients with *38-CpG AML Signature*[high] had significantly poorer OS (HR = 3.84; 95% CI = 3.01, 4.91; $P < 0.0001$) and EFS (HR = 2.22; 95% CI = 1.85, 2.65; $P < 0.0001$) in comparison to *38-CpG AML Signature*[low] group (Fig. 6e). In the AML02,08 test cohort, consistent results were observed, where patients within *38-CpG AML Signature*[high] group had significantly poorer OS (HR = 3.23; 95% CI = 1.75, 5.97; $P = 0.0002$) and EFS (HR = 3.22; 95% CI = 1.95, 5.33; $P < 0.0001$) as compared to the *38-CpG AML Signature*[low] group (Fig. 6f). Patient characteristics by *38-CpG AML Signature* groups in discovery and AML02,08 test cohorts are summarized in Supplementary Table 5 and show association with MRD1. Similar to results of the *AML Epigenomic Risk*, *38-CpG AML Signature*[high] was significantly associated with poor 5-year OS in independent NOPHO pediatric AML test cohort (HR = 2.40; 95% CI = 1.26, 4.57; $P = 0.0075$) and two adult datasets TCGA (HR = 1.72; 95% CI = 1.11, 2.67; $P = 0.0146$) and Beat AML (HR = 1.83; 95% CI = 1.17, 2.87; $P = 0.0081$) cohorts (Supplementary Fig. 5). In addition, *38-CpG AML Signature* stayed significant predictor of OS in discovery and AML02, AML08 test cohorts within MRD1 positive (discovery HR = 2.9; 95% CI = 1.91, 4.4; $P < 0.001$; AML02, AML08 test: HR = 2.38; 95% CI = 1.09, 5.19; $P = 0.029$) and MRD1 negative subgroups (discovery HR = 4.16; 95% CI = 2.95, 5.87; $P < 0.001$; AML02, AML08 test: HR = 4.57; 95% CI = 1.53, 13.68; $P < 0.001$) (Supplementary Fig. 6).

Multivariable analysis in the discovery cohort after adjusting for MRD1 status, risk group, FLT3 status, leucocyte counts at diagnosis, BM blast % at diagnosis, and age groups posed the signature as an independent predictor of OS (HR = 2.53; 95% CI = 1.86, 3.43; $P < 0.0001$) (Fig. 7a) and EFS (HR = 1.53; 95% CI = 1.23, 1.92; $P = 0.0002$) (Fig. 7b). Finally, multivariable analysis in the validation cohort after adjusting for the same confounding factors also validated the signature as an independent predictor of OS (HR = 2.34; 95% CI = 1.16, 4.70; $P = 0.017$) (Fig. 7c) and EFS (HR = 2.32; 95% CI = 1.31, 4.12; $P = 0.004$) (Fig. 7d).

Within the 229 standard risk group patients from the discovery cohort, patients with *38-CpG AML Signature*[high] had significantly poorer OS and EFS as compared to those with *38-CpG AML Signature*[low] (OS HR = 3.49; 95% CI = 2.00, 6.08; $P < 0.0001$; EFS HR = 1.46; 95% CI = 1.01, 2.11; $P = 0.0426$). Similar results were also observed in the 86 standard risk patients from the AML02,08 test cohort (OS HR = 3.08, 95% CI = 1.07, 8.89; $P = 0.0374$; EFS HR = 2.57; 95% CI = 1.14, 5.82; $P = 0.0233$). Kaplan-Meiers survival plots for EFS and OS by low, standard, and high-risk group categories are shown in Supplementary Fig. 7.

## AML Epigenomic Risk vs. 38-CpG AML Signature vs. standard of care

Here, we compared the predictive 5-year OS performance of *AML Epigenomic Risk*, *38-CpG AML Signature*, and clinical trial risk groups. In categorical (high-low risk) ROC analysis of the discovery cohort, the AUCs were 0.71, 0.69, and 0.67 for *AML Epigenomic Risk*, *38-CpG AML Signature*, and the risk group, respectively. In the AML02,08 test cohort, the AUCs were 0.71, 0.66, and 0.70. In NOPHO AML, the AUCs were 0.64 and 0.63 (risk group data not for NOPHO was not fully available) (Fig. 8a). Such a performance hierarchy is even more pronounced if models are used as continuous (probability of death at 5 years rather than high-low risk) (Fig. 8b). Pearson's correlation between *AML Epigenomic Risk* and *38-CpG AML Signature* was 0.7405 ($p < 0.0001$) in the discovery cohort, 0.6859 ($p < 0.0001$) in the AML02,08 validation cohort and 0.8065 ($p < 0.0001$) in the NOPHO AML cohort (Fig. 8c). These results were also summarized in confusion matrices, showing predicted vs. observed labels in discovery and the two validation cohorts for *AML Epigenomic Risk* model and *38-CpG AML Signature* (Fig. 8d). Overall, *AML Epigenomic Risk* is superior to *38-CpG AML Signature*, but it requires whole epigenome data (331556 CpGs) as compared to *38-CpG AML Signature*, which offers a simple and concise solution. Both models significantly improve upon current risk group classification for AML patients. (Fig. 9)

Given transcriptomic based leukemic stemness and drug resistance scores have been recently reported to have prognostic relevance[11,12], we compared the *AML Epigenome Risk* and *38-CpG AML Signature* in the context of these previously reported scores. EFS and OS by *AML Epigenome Risk* within low (Supplementary Fig. 8a) or high pLSC6 score groups (Supplementary Fig. 8b) and *38-CpG AML Signature* (Supplementary Fig. 8c, d), both epigenomic scores add additional predictive value beyond previously reported pLSC6 score[11,12]. Corresponding ROC curves in discovery and validation cohorts for current signatures in the context of previous transcriptomic signatures are also shown in Supplementary Fig. 8e, f, and further bootstrap-based Cox model for both cohorts are shown in Supplementary Fig. 8g, h.

## A long-read specimen-to-result protocol for epigenomic diagnosis and prognosis of AML

A cohort of leukemia patients from adult and pediatric hematology/oncology units from UF Health Shands Hospital was evaluated in-house using a rapid long-read nanopore sequencing protocol to assess the rigor of our proposed models prospectively from specimen collection to result generation. A total of 20 specimens from 17 patients of all ages (0.02–78 years) were considered for analysis for having a diagnosis of AML/MDS with > 1x WGS coverage, comprising 12 BM and 8 PB (3 patients had matched PB and BM). 10 of these patients were male and 7 were female. Time from specimen acquisition to sequencing start was 2 h. Basecalling, modcalling, and alignment was primarily completed in 48 h using up to 800 ng of input sheared DNA per PromethION flow cell per sample. Sequencing depth ranged from 1.14x to 24.62x, with a mean coverage of 12.96x (SD = 6.67). AML02 and

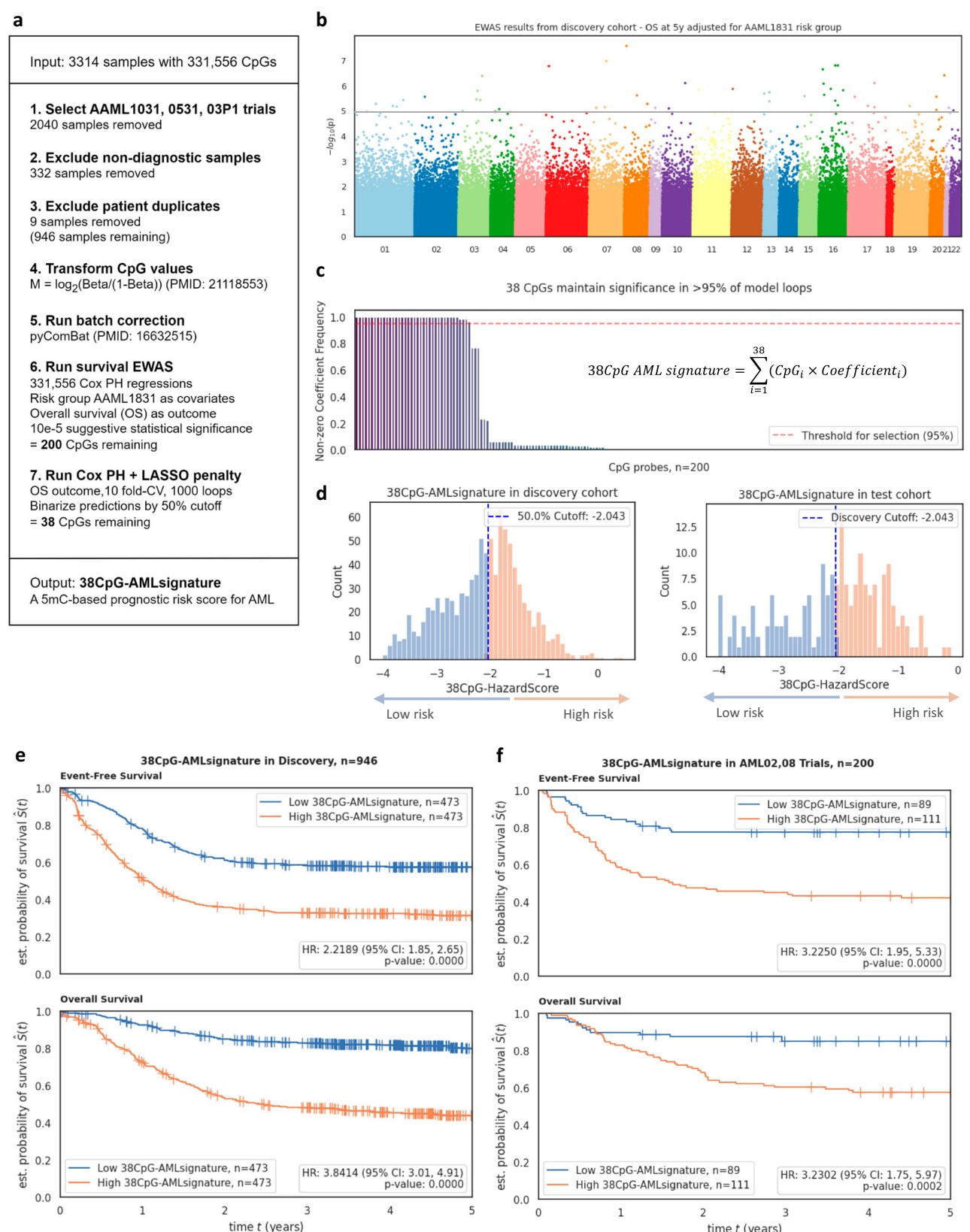

08 samples were available as purified DNA frozen for > 10 years, which, despite ideal nanodrop results, showed substantial pore blocking and a drop in yield likely due to DNA fragmentation. The remaining samples were processed as whole specimens and showed vastly superior yields. The mean read quality score (Q) was 24.94 (SD = 1.42), varying from a minimum of 21.9 to a maximum of 26.6, indicating consistently

reliable basecalling quality with *dorado = =0.9; dna_r10.4.1_-e8.2_400bps_sup@v5.0.0*. The median sample read N50 length was approximately 13.7 kb, with values ranging broadly from 4.3 kb to 25.9 kb, and a mean N50 of 14.2 kb (SD = 5.9 kb). Per sample results are described in Supplementary Table 6. 5mCG DNA modifications were collected for all sites in hg38 (~ 28x10e6 CpGs) but were subsequently

**Fig. 6 | Development and testing of 38CpG AML signature. a** Stepwise workflow for generating the *38CpG-AMLsignature* model, including data preprocessing, batch correction, survival analysis, and penalized Cox PH modeling, resulting in a concise 5mC-based prognostic risk score for AML. **b** Manhattan plot showing risk-adjusted EWAS results with the significance of CpG probes across the genome, highlighting 200 CpGs that remained significant after a multiple comparison threshold of $p < 10e-5$. **c** Stability analysis of 200 selected CpGs, showing the frequency of non-

zero coefficients across 1000 penalized model loops, which led to 38 CpGs being selected. **d** Distribution of 38CpG-HazardScores in the discovery and AML02,08 test cohort, dividing patients into high and low risk based on a 50% cutoff. **e** Kaplan-Meier survival curves for EFS and OS in the discovery cohort ($n = 946$) and (**f**) AML02,08 cohort ($n = 200$), comparing *38CpG-AMLsignature*high (orange) and *38CpG-AMLsignature*low (blue) groups. Hazard ratios derive from Cox PH regression with two-sided hypothesis tests. Individual n numbers are indicated in the figures.

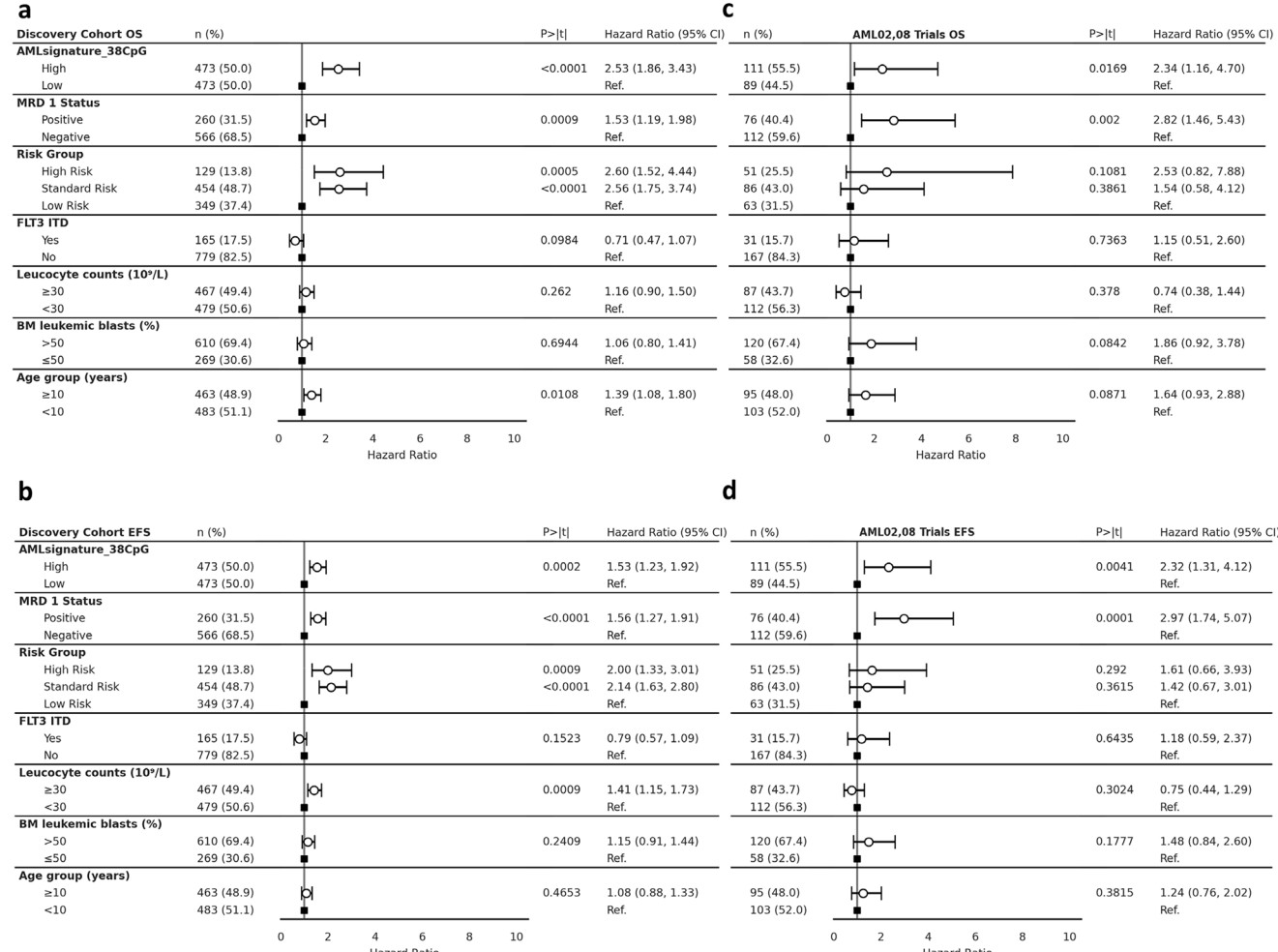

**Fig. 7 | Multivariate analysis adjusting for other confounding variables for association of 38 CpG signature with OS in (a)** discovery cohort (HR = 2.53; *P* < 0.0001) and (**c**) AML02,08 test cohort (HR = 2.34; *P* = 0.0169). Analysis for EFS was also performed in (**b**) the discovery cohort (HR = 1.53; *P* = 0.0002) and (**d**) AML02,08

test cohort (HR = 2.32; *P* = 0.0041). Hazard ratios derive from Cox PH regression with two-sided hypothesis tests. Individual n numbers are indicated in the figures. MRD1 minimal residual disease at end of first induction, FLT3 ITD FMS-like tyrosine kinase-3 internal tandem duplication, CI confidence interval, Ref. reference.

filtered to the genomic loci of ALMA (331556 CpGs). Any missing values were imputed using the corresponding CpG mean of the discovery cohort. Model parameters were then applied to the data, which mapped each sample onto ALMA. Finally, the *ALMA Subtype* model was applied, and if subtype prediction matched with AML or MDS with > 50% confidence, then *AML Epigenomic Risk and 38-CpG AML Signature* models were also applied and recorded.

**ALMA Subtype uncovers genomic lesions that govern leukemic phenotype**

We evaluated model performance across diverse clinical scenarios, including low blast count, relapse, MRD, low genomic coverage, rare leukemia subtypes, and diagnostically ambiguous cases. Individual results are detailed below.

Patient ALMA_1_BM (0–5 years, male; coverage 2.03x), was clinically diagnosed with acute monocytic leukemia (FAB M5, t(9;11)),

nanopore based methylation data predicted AML with t(8;16); KAT6A::CREBBP subtype and t(v;11q23); KMT2A-r, with KMT2Ar consistent with pathology diagnosis. This patient also had DNA methylation available from a 450 k Illumina array, and the predictions between nanopore and array platforms were consistent, suggesting no platform-specific bias. Patient ALMA_2_BM (5–13 years, male; coverage 6.52x), clinical report indicated AML t(8;21)(q22;q22), and DNA methylation demonstrated high-confidence concordance with AML subtype t(8;21); RUNX1::RUNX1T1 which was also confirmed with genomic sequence data (Fig. 6a). Similarly, ALMA_3_BM (13–39 years, female; coverage 1.14x), diagnosed with AML FAB M2, t(6;9), showed accurate prediction as AML with t(6;9); DEK::NUP214, supported by genomic confirmation of the NUP214 rearrangement at ultra-low coverage (Fig. 6b).

Patient ALMA_7_BM (60 + years, female; coverage 15.74x), residual AML with KMT2A rearrangement (10% blasts), was predicted as MDS-

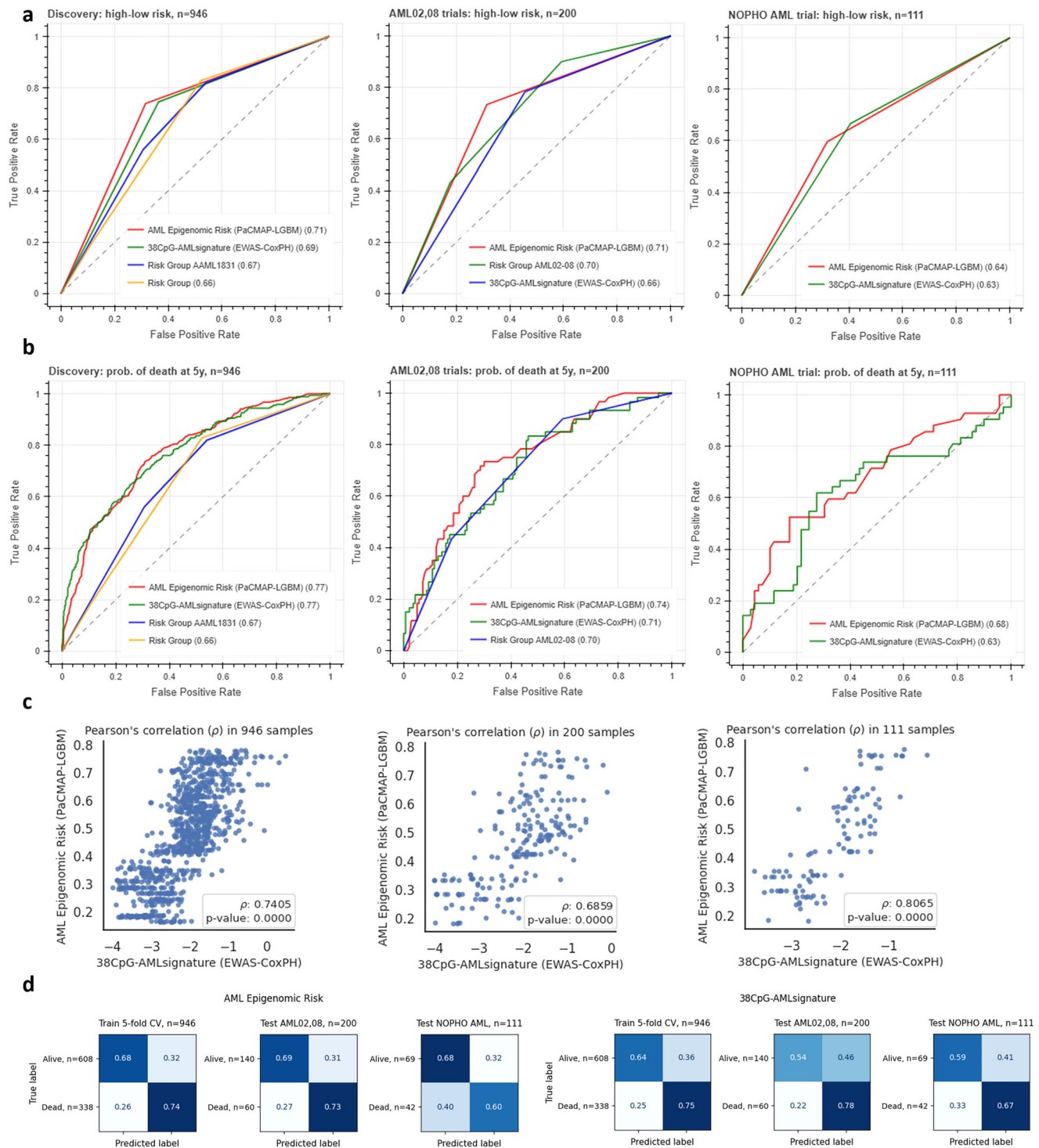

**Fig. 8 | AML Epigenomic Risk vs. 38-CpG AML Signature vs. standard of care.** ROC curves and AUC values for *AML Epigenomic Risk* model and *38-CpG AML Signature* as (**a**) categorical and (**b**) continuous variables in discovery, AML02,08 and NOPHO AML cohorts. **c** Pearson correlation scatterplot between *AML Epigenomic Risk* model and *38-CpG AML Signature* in discovery, AML02,08 and NOPHO AML cohorts. **d** Confusion matrices showing prediction rate for *AML Epigenomic Risk* and *38-CpG AML Signature*. Individual n numbers are indicated in the figures.

related AML but with a second likely call of AML with t(v;11q23); KMT2A-r, confirmed genomically by detection of KMT2A-CREBBP fusion. Patient ALMA_8_BM (39–60 years, male; coverage 7.77x), transitioning from MDS to AML (40% blasts, RUNX1T1 copy gain), was confidently classified as MDS-related secondary myeloid subtype, corroborated genomically by SF3B1, KRAS, and TP53 pathogenic mutations. Patient ALMA_9_BM (60 + years, male; coverage 18.12x),

diagnosed with MDS-NOS (trisomy 8, 15–20% blasts), was accurately predicted as MDS-related secondary myeloid, confirmed by RUNX1/RUNX1-AS1 mutations.

Patient ALMA_12_BM (60 + years, male; coverage 11.25x), with AML evolving from MDS, had a sample taken at day 68 s/p Vixeous and was correctly classified as "Otherwise-Normal Control" for having no evidence of disease. Patient ALMA_14_PB (39–60 years, female; coverage

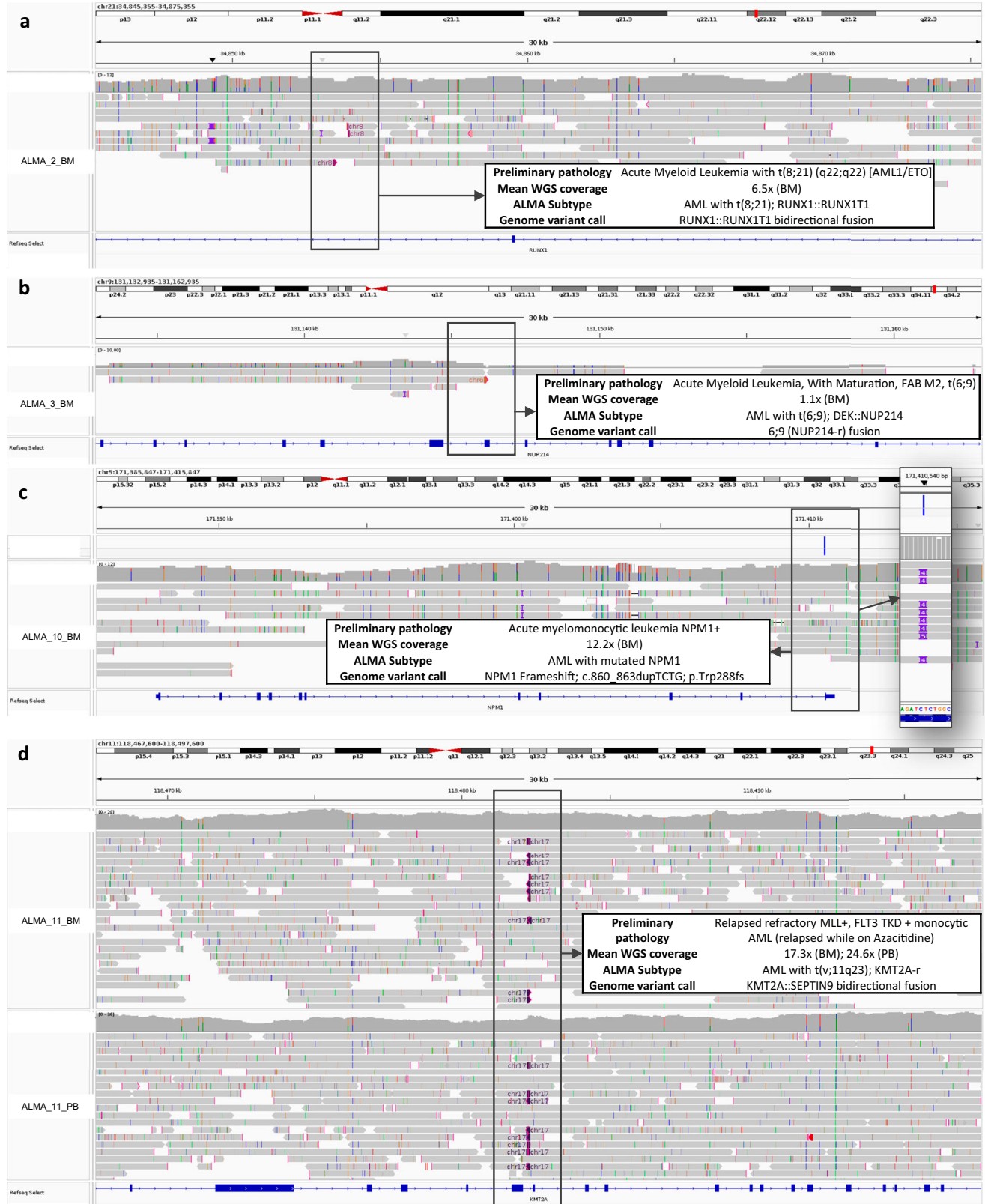

**Fig. 9 | Genomic confirmation of ALMA Subtype predictions in selected samples.** Integrated Genomics Viewer (IGV) plots display selected genomic alterations identified in the specimen-to-result testing cohort using nanopore based sequencing. All data were generated by PromethION 2 Solo sequencing, basecalled with dorado v0.9; dna_r10.4.1_e8.2_400bps_sup@v5.0.0 and aligned to hg38 using minimap2 v2.27. Each panel highlights key pathogenic variants correlated with clinical diagnosis and ALMA subtype prediction. **a** AML with t(8;21)(q22;q22)

(AML1-ETO); detection of RUNX1::RUNX1T1 bidirectional fusion (coverage: 6.5x, BM). **b** AML FAB M2 with t(6;9); DEK::NUP214 rearrangement confirmed (coverage: 1.1 x, BM). **c** Acute myelomonocytic leukemia NPM1 + ; frameshift insertion in NPM1 (c.860_863dupTCTG, p.Trp288fs) (coverage: 12.2 x, BM). **d** Relapsed refractory MLL +, FLT3 TKD + monocytic AML; identification of KMT2A::SEPTIN9 bidirectional fusion (coverage: BM 17.3 x, PB 24.6 ×). Each highlighted box indicates genomic variant regions and corresponding base-level details supporting the variant calls.

14.94x), with relapsed/refractory AML post-treatment, was accurately predicted as MDS-related secondary myeloid, confirmed genomically by detection of the DDX10-SKA3 fusion.

Patient ALMA_10_BM (60 + years, male; coverage 12.16x), clinically presenting acute myelomonocytic leukemia with NPM1 mutation, was predicted as AML with mutated NPM1, confirmed by genomic detection of a NPM1 pathogenic frameshift (c.860_863dupTCTG; p.Trp288fs) (Fig. 6c).

## Misclassified cases highlight current limitations of ALMA Subtype

Patient ALMA_6_BM (39–60 years, male; coverage 20.86x), presenting with relapsed AML (5% blasts), was classified as AML with mutated NPM1, despite genomic analysis revealing only benign NPM1 variants with the presence of KMT2A-KNL1 fusion. Although *ALMA Subtype* correctly identified the presence of AML at only 5% blasts and 90% cellularity, it misinterpreted the specific subtype. Our training cohort is composed primarily of pediatric patients for KMT2A-r (70.7% < 13 years; 2.3% ≥ 60 years) and older adults for NPM1-mutated cases (81.3% > 13 years; 27.1% ≥ 60 years), which may explain the misclassification for this case. Patient ALMA_4_PB (0–5 years, female; coverage 2.12x), presenting acute myelomonocytic leukemia (FAB M4), was inaccurately classified as MPAL T-Lymphoblastic/Myeloid subtype, with nanopore genomics showing FUS::ERG fusion presence. This patient had a uniquely complex karyotype: "46,XX,t(16;21)(p11.2;q22)[3] / 47,idem,der(2)t(2;12)(q33;q11.2),−12,+2mar[7] / 46,idem,der(2)t(1;2) (q23;q33)[5] / 46,idem,add(2)(q11.2),der(13)t(2;13)(q21;q14)[3] / 46,XX[2]". The correct clinical diagnosis would have been AML with FUS::ERG fusion, which is only present in 0.3–0.5% of AML cases[7]. Only 5 publicly available methylation patient samples representing this subtype exist, which limited its representation in our training cohort. These factors, in combination with 2.12x coverage, likely influenced the misclassification.

Patient ALMA_5_PB (0-5 years, male; coverage 11.46x), initially diagnosed with Down Syndrome AML, was predicted as AML with NUP98 fusion/MDS-related secondary myeloid, with trisomy 21 confirmed genomically but no detectable NUP98 fusion. The same was noted for patient ALMA_17_PB (0–5 years, female; coverage 19.36x), who also presented with Down syndrome AML and was predicted as AML with NUP98 fusion, despite genomic detection of trisomy 21 without evidence of a NUP98 fusion. Although the model successfully detected AML in both cases, it incorrectly assumed NUP98-r subtypes because no Down Syndrome AML cases are present in the training/discovery cohort. To our knowledge, no publicly available DNA methylation dataset of this important patient population exists.

## Paired PB and BM samples were correct and concordant despite tissue differences

Patient ALMA_11_BM and PB (13-39 years, male; coverage 17.30x BM, 24.62x PB), diagnosed with relapsed refractory MLL-positive AML, was accurately classified as AML with t(v;11q23); KMT2A-r in both tissues, corroborated by genomic detection of KMT2A-SEPTIN9 bidirectional fusion (Fig. 6d). Patient ALMA_13_BM and PB (60 + years, female; coverage 14.04x BM, 7.23x PB), presenting ambiguous flow cytometry suggesting biphenotypic leukemia (35% myeloblasts, dim CD19), was confidently predicted as MDS-related secondary myeloid in both specimens, supported genomically by RUNX1/RUNX1-AS1 mutations. Patient ALMA_16_BM and PB (60 + years, male; coverage 18.73x BM, 17.14x PB), diagnosed as AML with MDS-related changes, had samples taken status post Cytarabine and Daunorubicin therapy. For both tissues and in both genomics and epigenomics, no evidence of disease was detected, successfully predicting the remission status.

## Final ALMA dataset enables analysis of generative population models (GPM)

We employed GPM to demonstrate the statistical rigor of ALMA, as well as its potential for hypothesis-testing/generating research. First, certain joint distributions of our learned generative model were plotted against their empirical counterparts from the real dataset (Supplementary Fig. 9a). The inferred distributions qualitatively matched their empirical counterparts. Even after conditioning the model, joint distributions of variables agreed with their empirical counterparts (Supplementary Fig. 9b). For illustration, we have conditioned the learned model on race. As our inferred generative models are represented as probabilistic programs, we can transform and inspect them to search for complex nonlinear relationships among variables. For instance, we estimated mutual information between the dimensionality reduced methylome variables PaCMAP1-PaCMAP5 (Supplementary Fig. 9c). Finally, our probabilistic program representation also allowed for further analyses of mutual information that could potentially reveal causal patterns.

## Discussion

Genomic heterogeneity of myelogenous leukemia is proving to be far greater than the list of lesions currently present in diagnostic and prognostic guidelines. Currently, a third of patients at diagnosis are allocated to an ambiguous standard risk group, and a significant number in high- and low-risk are misplaced. These patients, if characterized adequately, could benefit from earlier transplantation, closer follow-up, and other treatment modalities. To address that, many studies identified DNA methylation and machine learning as promising avenues to improve diagnosis and prognosis in AML[4,13–16]. However, due to a lack of successful clinical implementation, these promising findings have yet to be practice-changing. Here, we unveil a specimen-to-result clinical workflow for diagnosis and prognosis of AML using epigenomics, unsupervised learning, and single-molecule nanopore sequencing.

We describe the development of ALMA, a publicly available map of acute leukemia heterogeneity created by compressing methylation levels of 331556 CpGs from 3314 high-quality patient samples. ALMA enabled the creation of *ALMA Subtype*, a supervised classifier of 27 clinical WHO 2022 diagnostic subtypes, plus otherwise-normal control. In addition, we develop two models of prognostic relevance that are predictive of clinical outcome: a global *AML Epigenomic Risk* model using ALMA that predicts risk of death within 5 years. In addition, by performing a risk-adjusted EWAS, we aimed to identify CpGs and genes that would not be already included in current risk classifications. These, we hypothesized, could offer complementary or novel roles in AML prognosis. An EWAS-CoxPH-based model named *38-CpG AML Signature* that predicts mortality within 5 years using DNA methylation levels of 38 CpGs.

Overall, the tools we present here (*ALMA Subtype, AML Epigenomic Risk,* and *38-CpG AML Signature*) hold high diagnostic and prognostic relevance. These are made accessible to clinicians by way of a specimen-to-result protocol that uses 220 μL of PB or BM to deliver simultaneous whole genome and epigenome sequencing, allowing for adjunct integration of our custom models with standard-of-care genomic assays.

As proof of concept, we applied the protocol to 20 patient samples from Hematology/Oncology adult and pediatric inpatient units. In this context, *ALMA Subtype* demonstrated strong clinical utility, accurately identifying various AML subtypes, including low-blast, MRD, relapse, and diagnostically ambiguous cases. High concordance between epigenomic predictions and genomic confirmation of known driver events (e.g., RUNX1::RUNX1T1, DEK::NUP214, NPM1 mutations, and KMT2A rearrangements) further validates its diagnostic precision. Notably, paired PB and BM analyses yielded concordant results,

suggesting robustness of epigenomic profiling regardless of specimen type, enhancing flexibility in clinical applications.

However, notable limitations emerged, particularly in cases with rare or poorly represented subtypes within the training cohort. Misclassification incidents involving FUS::ERG fusions, KMT2A rearrangements in older adults, and Down Syndrome-associated AML highlight critical gaps in training data availability. The small representation of these subtypes, particularly Down Syndrome AML and ultra-rare genomic fusions, limits the model's generalizability, highlighting the urgent need for publicly available methylation data representing these important patient populations.

In cases of misclassification, coverage levels, though generally adequate, played a contributory role. Ultra-low coverage (1-2x) appears feasible for detecting hallmark epigenomic signatures in some cases (e.g., DEK::NUP214); however, accuracy declines for subtypes poorly represented in training data. Ensuring robust representation of all AML subtypes of all patient ages and adopting complementary genomic validation in ambiguous scenarios are key to clinical accuracy.

Biologically, our efforts reveal that driving structural variations or recurrent mutations elicit a distinct epigenomic profile that matches closely with diagnostic subtyping and risk stratification. Regardless of the extent of the genomic insult − from single-nucleotide variants to major structural variants − the resulting phenotype may be unambiguously identifiable at the epigenomic layer. Such findings have been described in other domains: A seminal study conducted in central nervous system tumors combined DNA methylation array data with machine learning tools to conclusively classify patients who might otherwise be misdiagnosed by standard of care methods[5,17]. These findings were clinically validated and shown to impact neurosurgical strategy[18]. Though these techniques have been successfully applied to other malignancies like sarcomas[6], they have yet to be comprehensively described in the hematopoietic system.

From a computational biology perspective, we aimed to set a high standard of transparency and rigor by creating a step-by-step electronic notebook with source code, explanation, and resulting figures for all analyses undertaken in this study (https://f-marchi.github.io/ALMA/). In addition, we release here an interactive interface for ALMA and a Python package with our three models (https://github.com/f-marchi/ALMA-classifier). These are meant to be open-source tools that evolve as new data and algorithms are released, paving the way for more affordable advanced molecular profiling of patients regardless of geography.

Given that prognosis is a function of diagnostic subtype and standard of care, the clusters present in ALMA also translate to prognostic classification, evidenced by the ability of *AML Epigenomic Risk* to serve as a substantial predictor of overall survival with robust testing results in fully independent trials. This is not only due to the algorithms and datasets used, but also due to the stable nature of DNA methylation, which is covalently bound to double-stranded DNA and hence a remarkably inert marker of cellular differentiation and tissue of origin. Two prognostic models were created because testing for a limited number of CpGs may allow for broader utility where whole-genome sequencing is unavailable. Our EWAS-CoxPH based stepwise approach resulted in a *38-CpG AML Signature* with similar predictive capacity as *AML Epigenome Risk*, allowing centers and investigators to choose between the two options according to their resources.

Out of the 38 CpGs identified in the signature through risk-adjusted EWAS, 30 mapped to genes potentially involved in AML prognosis. Several of these genes have established roles in hematological malignancies, highlighting their biological relevance. For instance, TRAF7 (cg02678414) acts as a tumor suppressor, and its restoration inhibits AML proliferation via the KLF2-PFKFB3 glycolytic pathway[19–21]. Similarly, CD93 (cg14928764) marks leukemic stem cells in MLL-rearranged AML and a therapy-resistant quiescent population

in CML, while its inhibition reduces AML proliferation[22–24]. VNN1 (cg01052291), although not directly linked to AML previously, is implicated in oxidative stress and immune regulation in chronic ITP, suggesting a potential role in leukemia-related inflammation or oxidative stress mechanisms[25–27]. EXT1 (cg02905663), a regulator of apoptosis and heparan sulfate biosynthesis, functions as a tumor suppressor in ALL through ERK1/2 signaling modulation and interacts with NOTCH1-FBXW7 pathways[28,29]. Conversely, EXT1 knockdown induces apoptosis in multiple myeloma cells, underscoring context-dependent roles in hematologic malignancies[30]. Finally, P2RX4 (cg19357999) is upregulated in relapsed acute leukemia, promoting plasma cell survival through purinergic signaling[31–33]. The presence of these biologically relevant genes in our signature reinforces the validity of this approach in identifying potentially prognostic methylation markers that complement existing risk classifications in AML.

Long-read sequencing was chosen for clinical implementation for being the only technology presently capable of detecting most structural variants, repeats, and DNA modifications out of a single run. Furthermore, 5mCG calling is done natively at a single-molecule level without bisulfite conversion and amplification bias. Knowing these variants is a fundamental requirement for a comprehensive understanding of a patient's leukemia genome, which escapes the capacity of short read sequencing, PCR, and fluorescence-based tools. However, costs per genome are still prohibitive, which begs for the development of more competitive and innovative platforms and strategies. While the findings of this study are promising, further research in larger cohorts is required to establish a benchmark for the minimal threshold of leukemic blasts required for reliable classification via long-read sequencing. Further exploration of genetic markers and their roles in refining risk assessment will also be crucial to enhance predictive accuracy.

In conclusion, this study introduces a clinical workflow that utilizes epigenomic sequencing for sensitive and specific diagnosis and prognosis of AML patients. We aimed to map the heterogeneous landscape of acute leukemias by way of the unsupervised ALMA foundational model, which has been structured as an ever-evolving open-source toolkit. And finally, we introduced three robust algorithms: *ALMA Subtype*, *AML Epigenomic Risk*, and *38-CpG AML Signature*, all of which showed state-of-the-art diagnostic or prognostic capacity, respectively, and were made available through the Python package *alma-classifier* released with his study. These efforts highlight the potential for increased affordability, speed, and accuracy in oncology care and trial design, ultimately benefiting patients worldwide.

## Methods

### Patient characteristics

This research complies with all relevant ethical regulations. Written informed consent was obtained from all participants prior to sample collection. The study protocols were approved by the University of Florida's Institutional Review Board.

In this study, we first assembled and harmonized raw data from publicly available methylation datasets of acute leukemias from 11 clinical trials/studies, making it one of the largest cohorts to be evaluated. Supplementary Table 1 provides details on these 11 cohorts: NOPHO ALL92-2000 ($n = 933$)[34], AAML0531 ($n = 628$)[35,36], AAML1031 ($n = 587$)[37,38], BeatAML ($n = 316$)[16], TCGA AML ($n = 194$)[39], French GRAALL 2003-2005 ($n = 141$)[40], TARGET ALL ($n = 131$)[41], CETLAM SMD-09 ($n = 166$)[42], AAML03P1 ($n = 72$)[43], Japanese AML05 ($n = 64$)[15], and CCG2961 ($n = 41$)[44], resulting in a total of 3314 patients after preprocessing and quality control assessment. Samples were obtained either from bone marrow or peripheral blood at diagnosis, relapse, or remission. DNA methylation data was procured using the Illumina methylation array 450 k or EPIC array, which share 452,453 probes with the same chemistry and design.

For the development of *ALMA Subtype*, our fine-tuned supervised diagnostic model, a subset of 2471 samples were selected for having WHO 2022 diagnostic annotation data (Supplementary Table 2). Clinical annotations such as karyotypes, cytogenetics, gene fusion, or otherwise-specified diagnostic information were used to create European LeukemiaNet (ELN) and World Health Organization (WHO) 2022 subtypes.

For the development of both the models of prognostic relevance, we restricted our analysis to specimens obtained at diagnosis (predominantly bone marrow aspirates) from the Children's Oncology Group (COG) trials AAML1031 (NCT01371981), AAML0531 (NCT00372593), and AAML03P1 (NCT00070174) through GSE190931, GSE124413, and GDC_TARGET-AML datasets, respectively. The clinical endpoints were defined as: (i) Minimal residual disease after induction 1 (MRD1): positive MRD1 if ≥1 leukemic cell per 1000 mononuclear bone marrow cells (≥ 0.1%) determined by flow cytometry; (ii) event-free survival (EFS) defined as the time from study enrollment to induction failure, relapse, secondary malignancy, death, with event-free patients censored on last follow-up; (iii) overall survival (OS) defined as the time from study enrollment to death, with living patients censored on the date of last follow-up.

To independently test the findings derived from the discovery cohort, we processed in parallel DNA methylation data from 200 AML patients treated on the multi-center clinical trials AML02 (NCT00136084) and AML08 (NCT00703820)[45,46]. For further validation, we provided our models to collaborators via the open-source python package created for this study (*alma-classifier*), which enabled evaluation of our model performance in an external independent cohort of 142 patient samples who were treated according to NOPHO93 and NOPHO2004 protocols, as previously described by Krali et al. [4]. Out of the 142 samples, 76 had matching WHO 2022 diagnosis and 111 had prognostic model predictions available. The 31 samples that did not have prognostic predictions were either called not confident by *ALMA Subtype*, or were predicted to be APL, ALL, MPAL.

## Data quality control and preprocessing

SeSAMe[47], which is the selected methylation processing software of the National Cancer Institute's Genomic Data Commons and COG[28] was utilized for preprocessing of the raw methylation array[38]. Preliminary quality control exclusion criteria included: (i) 12003 sex-linked and non-CpGs; (ii) 47382 CpGs deemed unreliable based on literature benchmarking[48]; (iii) 460 samples due to Illumina quality control p-value failure; (iv) 61512 CpGs with over 5% missing values; (v) 60 non-hematopoietic samples; (vi) 11 samples considered outliers according to PCA analysis. Finally, interpolation by batch mean filled the remaining missing values[49]. To adjust for potential confounding variability, batch correction was performed using *ComBat*[50,51]. The final dataset of 331556 CpGs and 3314 samples was considered for downstream statistical analyses (Supplementary Fig. 10).

## Development of custom models of diagnostic and prognostic relevance

**Acute leukemia methylome atlas (ALMA).** To capture the heterogeneous landscape of acute leukemia by epigenomics, we used a dimensionality reduction algorithm called Pairwise Controlled Manifold Approximation (PaCMAP)[52], which allowed compression of processed CpG β-values into two dimensions for visualization and five dimensions for supervised classification analysis.

**ALMA Subtype.** To devise the diagnostic classifier, we used five PaCMAP dimensions (coordinates), which represent the combined compressed values for 331556 genome-wide CpGs. These were input to Light Gradient Boosting Machine (LGBM), a decision-tree-based supervised learning algorithm. Hyperparameter tuning using 5-fold cross-validation (CV) involved Lasso and Ridge penalties with balanced class weights to account for rarer subtypes. For diagnosis, we assessed accuracy per class of 27 WHO subtype categories plus otherwise-normal control (a one-vs-rest multi-class classifier of 28 clinical subtypes). Not all samples, however, had the WHO 2022 annotation available in the dataset to serve as ground truth, so only those with annotations were used (*n* = 2471 discovery cohort; *n* = 104 validation cohort). The *otherwise-normal control* category is composed of samples described as "normal" or believed to be without disease. These are normal bone marrow or peripheral blood samples from either otherwise-healthy subjects or patients in remission and without evidence of disease. No cell lines or sorted primary cells were used as controls.

**AML Epigenomic Risk.** To devise the prognostic classifier, 946 AML-only PB or BM samples at diagnosis were used. These were selected from the Children's Oncology Group (COG) trials AAML1031 (NCT01371981), AAML0531 (NCT00372593), and AAML03P1 (NCT00070174) through GSE190931, GSE124413, and GDC_TARGET-AML datasets, respectively. The same strategy listed as above was used, where five PaCMAP dimensions were input to an LGBM classifier with hyperparameter tuning in 5-fold CV. The target output, however, was OS at 5 years (dead/alive status). Testing was performed in an independent dataset of 200 AML patients from multi-center AML02 and AML08 trials, as well as 111 patients from the NOPHO AML cohort.

**38-CpG AML Signature.** As an alternative approach and to devise a concise epigenomic signature of prognostic relevance, we used the same samples as described above. Using 331556 CpGs, we conducted an epigenome-wide association study (EWAS) using Cox Proportional Hazards regression to identify CpGs with methylation levels most predictive of OS at 5 years while adjusting for risk group categories. Next, we selected CpGs at a suggested *p*-value threshold of 10e-5 as input to the LASSO-based multivariate Cox-PH model, with 1000 iterations of 10-fold cross-validation. CpGs selected in ≥95% of the models were utilized to create an epigenomic signature. To create a binary risk score, patients were further categorized into either *38-CpG AML Signature*^low or *38-CpG AML Signature*^high groups by median cutoff.

## Main software

The methylation array preprocessing software used was *methylprep* v.1.7.1 in Python 3.7, followed by *methylcheck* v0.8.5 in Python 3.8. Statistical learning analyses were implemented with *scikit-learn* v1.2.2, *scikit-survival* v0.20.0, *statsmodels* v0.13.5, and *lifelines* v0.27.7[53–55]. Patient characteristics table used *tableone* v0.7.12[56]. Plotting packages were *matplotlib* v3.7.1, *bokeh* v3.3.4, and *seaborn* v0.12.2[57–59]. Data structure and manipulation packages were *numpy* v1.24.4 and *pandas* v2.0.3[60,61]. Sankey plots were adapted from *pySankey* v0.0.1. Finally, *pacmap* v0.7.0 and *lightgbm* v3.3.5 were used[62].

## Machine learning rigor, clarity, and reproducibility

To abide by the criteria recently proposed in the literature of machine learning applications in life sciences and establish the rigor of our bioinformatic pipelines[63], we made the raw training data, processed training data, model weights, and source code publicly available and open source. All methods and results from this study were written in a step-by-step format using Jupyter Book through GitHub Pages. In addition, testing samples come from independently conducted clinical trials and independently processed pipelines. Specimen-to-result testing was done in-house using nanopore sequencing to validate findings using distinct platform chemistries.

## Specimen-to-result testing protocol

**Puregene DNA extraction.** Whole blood or bone marrow (220 μL) samples in EDTA were processed using a modified QIAGENE Puregene

Blood Core Kit protocol adapted from Goenka et al.[64]. Briefly, RBC lysis in clot-free samples followed by centrifugation at 16,000 x g for 1 min was used to pellet white blood cells. Cell lysis was achieved by adding 600 μL of Cell Lysis Solution, pipetting, and vortexing at 3000 rpm for 20 s. RNase A Solution (1.5 μL) was added, and the mixture was inverted 25 times before incubation at 56 °C for 5 minutes. Protein precipitation was facilitated by adding 200 μL of Protein Precipitation Solution, followed by incubation on ice for 5 minutes and centrifugation at 16,000 x g at 4 °C for 4 min. The supernatant was transferred to a tube containing 600 μL of isopropanol, gently flipped 50 times, and centrifuged at 16,000 x g for 1 min to pellet the DNA. The DNA pellet was washed with 600 μL of 70% ethanol, centrifuged, and dried at 37 °C for 2 min before being dissolved in 100 μL of DNA Hydration Solution and incubated at 65 °C for 10 min.

**QIAamp Automated DNA extraction.** As an alternative method to purifying DNA that proved equally effective but more practical, the QIAamp DNA Blood Mini Kit was used in a QIAcube automated robot according to manufacturer protocol. Using this method, the time for automated 2-sample processing was 37 min, which enabled us to prepare library prep steps in the meantime, allowing for a 2 h specimen-to-sequencing-start time.

**DNA shearing and quantification.** DNA samples were sheared using a 30 G 0.5in needle attached to a 1 mL Luer slip tip syringe. The needle was placed at the bottom of a 1.5 μL DNA LoBind sample tube, and the liquid was drawn up to the 0.3 mL line and expelled back into the tube. This shearing process was repeated five times to achieve the N50 of 10 kb. For DNA quantification, the Qubit dsDNA BR or HS Assay was used according to the manufacturer's protocol in triplicate measurements.

**Library preparation and sequencing.** For sequencing, the Rapid Sequencing Kit V14 - gDNA (SQK-RAD114) was used as per manufacturers protocol with the following important modifications: (i) input mass was optimized to 800 ng; (ii) volume of fragmentation mix (FRA) and rapid adapter (RAP) doubled to 2 μL. The prepared library was loaded into a PromethION 2 Solo sequencer by slowly rotating the pipette plunge.

**Basecalling, modcalling, and alignment.** Raw signal data (POD5) collection was done using *MinKNOW* software and transferred. Basecalling was executed using Oxford Nanopore Technology (ONT) basecaller *dorado* v0.9.1 using super high accuracy (sup) model version 5.0.0. The GPU used was one local NVIDIA GeForce RTX 4090. The reference genome used was *GCA_000001405.15_GRCh38_no_alt_analysis_set.fna* (hg38). Alignment was conducted using *minimap2* using the *lr:hq* preset[65]. 5-methylcytosine-guanine (5mCG) modifications were called by ONT modcaller *remora*. Both mod calls and alignment were done through *dorado* command line.

**Long-read data processing.** Post-sequencing data processing included sorting BAM files using *samtools* v1.13[66]. The total number of reads, unmapped reads, total bases, and N50 values were reported alongside mean coverage on hg38 and median accuracy scores. Genomic variant analysis was done using *epi2me desktop* v5.2 and *wf-human-variation* v2.6.0, which covered the following modalities and tools: single-nucleotide variants (SNVs), structural variations (SVs), copy number variations (CNVs), and short tandem repeats (STRs)[67]. SNVs were identified with *Clair3* v1.0 and classified into pathogenic SNVs and drug response SNVs by *ClinVar*[68]. SVs, including inversions, duplications, and translocations, were detected using *Sniffles 2* v2.0.7-epi2me using "−non-germline" parameter[69]. CNVs were called with *QDNAseq* v1.34.0 and/or *Specter* v0.2.2, specifying abnormalities such as gains and losses of chromosomes or chromosomal segments[70]. STRs were identified

using *straglr-genotype* v1.4[71]. Finally, 5mC and 5hmC modification tags were processed into bed files with *modkit* v0.4.3. Specific commands are available in the electronic notebook under the nanopore chapter.

**Model testing on long-read DNA methylation data.** To test models using the nanopore platform, we created a *pacmap_reference* BED file containing genome coordinates for 331556 CpGs according to hg38. For each nanopore sample, a BED file provided by *modkit* methylation levels for 28 M CpGs is loaded and merged with the *pacmap_reference*. DNA-methylation data structure is the same between Illumina arrays and nanopore sequencing since it represents the fraction of methyl groups encountered at a given CG locus, so no data transformation was necessary. The pipeline then proceeds as described in the code availability section.

### ALMA generative population models

We aimed to validate ALMA using the framework of GPMs, which automate error-prone aspects of data cleaning and analysis. Briefly, a GPM consumes a set of sparsely overlapping datasets and infers through Bayesian structure learning[72] a probabilistic program representing a generative model of the combined data (Supplementary Fig. 11). This enables interaction with the learned generative program via natural language prompts semantically parsed and translated into queries in GenSQL[73] and optimized and lowered to probabilistic programs in SPPL[74]. Moreover, the learned generative programs can generate synthetic datasets that match the statistical properties of the combined datasets while not revealing any sensitive patient information or suffering from missing values (Supplementary Fig. 9a, b). Further, the probabilistic program representation facilitates causal analyses or poststratification (Supplementary Fig. 9c)[75].

### Reporting summary

Further information on research design is available in the Nature Portfolio Reporting Summary linked to this article.

## Data availability

Discovery (training) raw DNA methylation array data analyzed in this study were obtained from Gene Expression Omnibus (GEO) under accession codes GSE190931[37,38], GSE124413[35,36], GSE133986[15,76], GSE159907[16,77], GSE152710[42], GSE49031[34,78], GSE147667[40], as well as from Genomic Data Commons (GDC; https://portal.gdc.cancer.gov/) under categories GDC-TARGET-AML[35,43,44,79], GDC-TCGA-AML[80], GDC-TARGET-ALL[41]. Processed, patient-level methylation and clinical data from the discovery/training cohort are available at https://doi.org/10.5281/zenodo.15653263[81] through https://github.com/f-marchi/ALMA/releases/tag/v0.2.0 and Source Data, respectively. The raw nanopore genome sequencing data generated in this study are not publicly available because participant consent did not explicitly cover deposition in public repositories. However, participants did consent to the use for research purposes, and the data are available for bona fide research upon reasonable request to the corresponding author (jatinderklamba@ufl.edu). Access is subject to approval by the relevant institutional ethics board and completion of a Data Use Agreement (DUA) to ensure compliance with ethical and legal obligations. Processed, de-identified methylation and clinical data are available at the same DOI and in Supplementary Table 6, respectively. The remaining data are available within the Article, Supplementary Information, or Source Data file. Source data are provided in this paper.

## Code availability

Source code for how data was collected, as well as for all analyses, figures, and tables, are publicly available at the electronic notebook we created for this study: https://f-marchi.github.io/ALMA/ and as source code (https://github.com/f-marchi/ALMA/releases/tag/v0.2.0; https://doi.org/10.5281/zenodo.15653263[81]). Software and hardware

information for all analyses are noted at the end of each chapter under the section "Watermark". We are also releasing *alma-classifier* v0.1.4 (https://github.com/f-marchi/ALMA-classifier; https://doi.org/10.5281/zenodo.15636415[82]), enabling others to use, reproduce, and build upon the three models described here. At release (v0.1.4), it is structured as an open-source Python package and Docker image that takes as input methylation values from 331556 CpGs (with some room for missing values) and outputs calculated predictions for *ALMA Subtype*, *AML Epigenomic Risk*, and *38-CpG AML Signature*.

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

## Acknowledgements

We thank the patients and families who participated in the studies, especially those who have faced the greatest peril in the process. Without their generous contributions, this research would not have been possible. We gratefully acknowledge the support and collaboration of all research centers and clinicians involved in the collection of data. Special thanks to Luis Salazar, PhD, from Oxford Nanopore for his insightful guidance, Pedro Orsini for the invaluable clinical discussions, and Joao Loula for fundamental contributions to both algorithmic and software infrastructure of generative population models. Finally, we acknowledge the use of LLMs for Python code troubleshooting, function design/labeling, and commit message autogeneration. Research

reported in this publication was supported by the NIH/NCI (R01CA132946), American Cancer Society/St Baldrick's Foundation (SAP 21-061-01), UF Health Cancer Center, supported in part by state appropriations provided in Fla. Stat. § 381.915 and the NIH/NCI P30CA247796. NIH awards U10CA180886, U10CA98413, and U10CA098543 supported the COG clinical trial. NCTN Statistical Data Center supports U10CA180899. The content is solely the responsibility of the authors and does not necessarily represent the official views of the National Institutes of Health or the State of Florida.

## Author contributions

F.M. and J.K.L. contributed to the conception and design of the study. Study materials, data collection, and assembly were provided by F.M., V.M.S., A.Ö., M.L., A.K.S., R.C.R., A.G., R.A., J.R., C.R.C., S.P., R.R., S.M., E.A.K., T.A.A., J.N., and J.K.L. Data analysis and interpretation were conducted by F.M., M.L., F.S., M.G., A.E., V.K.M., O.K., T.J.T., X.C., T.A.A., S.M., S.P., R.J.M., N.H.K.N., P.S., W.S., and J.K.L. Manuscript writing was performed by F.M., A.Ö., M.L., and J.K.L. All authors reviewed and provided final approval of the manuscript.

## Competing interests

F.M. and J.K.L. are inventors on an international patent application related to this work (PCT/US2024/058595), filed on December 5, 2024, by the University of Florida Research Foundation, Inc. The authors declare no other competing interests.
