## [Transparent Peer Review file · Nature Communications]

Epigenomic diagnosis and prognosis of Acute Myeloid Leukemia

Corresponding Author: Dr Jatinder Lamba

Version 0:

Reviewer comments:

Reviewer #1

(Remarks to the Author)

The manuscript, "Long-read epigenomic diagnosis and prognosis of Acute Myeloid Leukemia," introduces the Acute Leukemia Methylome Atlas (ALMA), a diagnostic model integrating harmonized DNA methylation data from 3,314 patients. ALMA predicts 27 WHO 2022 acute leukemia subtypes with high accuracy (96.3% in discovery and 90.1% in validation cohorts). Two prognostic tools were developed: the AML Epigenomic Risk model, leveraging whole epigenome data for survival prediction, and the 38CpG AML Signature, a concise classifier based on 38 CpG sites.

The study also presents a rapid, long-read nanopore sequencing workflow enabling simultaneous genomic and epigenomic analysis, achieving accurate diagnosis and prognosis predictions within 24 hours. These findings highlight the clinical potential of epigenomic profiling for improved AML subtype stratification and personalized therapeutic approaches.

This study is a comprehensive and generally high-quality study. Nevertheless, there are some points of criticism and concerns that should be addressed by the authors.

- The study does not clearly address the differences between peripheral blood (PB) and bone marrow (BM) samples and how variations in blast count may impact the quantitative DNA methylation values and the resulting signatures.
- The validation cohort for the ALMA 5-dimensional model is relatively small, which may undermine the robustness and generalizability of the findings.
- ALMA shows promise in stratifying subtypes, but the authors do not adequately address potential challenges in applying it to rare subtypes or ambiguous cases.
- Figure referencing appears to be incorrect, particularly for WHO discovery/validation data in Figures 2A/C and unknown discovery/validation data in Figures 2B/D.
- While the study uses a large dataset from multiple cohorts, it is unclear how well it captures the diversity of AML subtypes and patient demographics. Potential sampling biases, such as geographic, age, or ethnic representation, should be considered.
- Metrics such as macro and weighted F1 scores show significant differences between discovery and validation cohorts. These discrepancies are not sufficiently explained.
- The potential for overfitting is not discussed, despite the high accuracy metrics observed in the discovery phase.
- The prognostic model is evaluated only in the pediatric AML sub-cohort. The authors should explain whether the findings hold true for adult AML populations.
- The association between achieving MRD1 and genetic risk is not explored. The comparison between epigenomic and genetic models should be clarified, particularly in how risk groups are defined.
- For the 38CpG AML signature, the authors do not discuss whether adding or removing CpGs could replicate the performance of classical risk models.
- The choice of CpGs for the 38CpG AML signature was based on a p-value threshold of $10e-5$, which could be arbitrary and exclude biologically relevant CpGs.
- While the LightGBM model is efficient, it is unclear whether the authors compared its performance to other methods, such as deep learning, which might better capture complex patterns.
- The link between the CpGs in the 38CpG AML signature and underlying genes or biological mechanisms is not sufficiently addressed.
- The authors do not indicate whether statistical significance testing was performed to compare their proposed models with existing standards. The models predict survival at 5 years, but the outcomes are not compared with other established

prognostic factors, such as FLT3 and NPM1 mutations. Integrating these molecular features could enhance predictive accuracy.

- The manuscript does not provide sufficient details about the sequencing coverage/depth or genetic findings derived from Nanopore sequencing.
- The authors do not evaluate whether Nanopore sequencing might introduce errors in methylation calling due to technical limitations.
- Imputation of missing values using the mean CpG values could introduce bias, particularly for rare subtypes or outliers.
- The Nanopore analysis is based on very few samples. The cohort size should be increased, and patient follow-up data should be included to evaluate prognostic accuracy.
- Claims of a "close match" between model predictions and pathology are questionable. For example, AML with MDS changes maps to "otherwise normal control," and biphenotypic AML maps to "MDS-related."
- The language of the manuscript is generally strong and concise; however, minor adjustments are needed:
- The title is somewhat misleading, as Nanopore longread sequencing makes up a rather small part of the manuscript.
- The manuscript does not adequately discuss limitations, including:
 - o The small cohort size for rare subtypes.
 - o Potential biases in methylation array data.
 - o Prognostic data limited to the pediatric cohort.

(Remarks on code availability)

Reviewer #3

(Remarks to the Author)

Evaluation Report for Manuscript "Long-read Epigenomic Diagnosis and Prognosis of Acute Myeloid Leukemia"

Summary of the Manuscript:

This publication examines significant obstacles in the diagnosis and prognosis of Acute Myeloid Leukemia (AML), a condition for which existing clinical techniques frequently lack precision and scalability. The authors propose a paradigm shift by integrating long-read sequencing technologies with DNA methylation profiling to address limitations in current methodologies, including incomplete diagnostic coverage, significant variability across datasets, and restricted interpretability of machine learning models. Although the study is unique and methodologically sound, it is crucial to identify aspects that necessitate further elucidation or enhancement to optimize its clinical and translational significance. The article introduces a novel method to enhance the diagnostic and prognostic framework of Acute Myeloid Leukemia (AML) by utilization of long-read sequencing technology and DNA methylation profiling. The Acute Leukemia Methylome Atlas (ALMA) is a comprehensive DNA methylation database that accurately distinguishes AML subtypes, achieving 96.3% in the discovery cohort and 90.1% in the validation cohort. Innovative prognostic models, such as AML Epigenomic Risk and a 38-CpG signature, demonstrate robust predictive accuracy for overall survival (OS) and event-free survival (EFS). Comprehensive whole-genome and epigenome sequencing were amalgamated utilizing long-read technologies and validated across 3,314 patients from 11 heterogeneous cohorts. The study emphasizes the value of DNA methylation profiling in improving AML diagnoses and prognostics, providing a more accurate and scalable framework for precision oncology. The open-source web-based tool associated with ALMA also supports clinical and scientific applications, hence emphasizing the study's practical significance.

Major Concerns:

1. Although the models were verified using retrospective datasets, their efficacy in prospective, real-world multicenter studies has yet to be evaluated. This stage is essential for facilitating clinical adoption. The study fails to address the representativeness of the datasets about global AML populations, especially in low-resource or underrepresented contexts. Mitigating these constraints would strengthen the validity of the conclusions.
2. The implementation of long-read sequencing in standard diagnostics may encounter obstacles related to cost, infrastructure, and technical knowledge. A cost-benefit analysis and a comparison with current diagnostic techniques would enhance the discourse. Regulatory procedures for sophisticated sequencing diagnostics must be evaluated, including potential barriers to approval or harmonization.
3. Despite their efficacy, techniques such as PaCMAP and LightGBM frequently face criticism for being "black box" models. Incorporating a discourse on feature significance or interpretable results may reconcile the disparity between computational intricacy and therapeutic relevance.
4. The diagnostic framework's elevated sensitivity may result in overdiagnosis or misclassification, especially in borderline instances. Implementing protections or rules would alleviate this risk for the document.
5. Although hazard ratios (HRs) for survival are persuasive, supplementary performance measures, including sensitivity, specificity, and predictive value, must be presented to comprehensively evaluate the clinical relevance of the prognostic models. An elucidation of how the 38-CpG signature correlates with or enhances established prognostic indicators such as FLT3-ITD or NPM1 mutations would contextualize its significance.
6. Selection of Features for 38-CpG Signature. The article outlines the EWAS-Cox PH-LASSO pipeline but fails to elucidate the criteria for selecting the final set of CpGs and whether any bias may have affected this decision. The text lacks adequate information regarding the statistical criteria for significance, such as p-values for CpG inclusion. A clear justification for these thresholds and their biological importance is required. As well as for the data harmonization. Although the publication amalgamates data from 11 cohorts, it fails to address the mitigation of batch effects or discrepancies between platforms (e.g.,

- Illumina EPIC versus 450K arrays). This is essential for guaranteeing dependable model training and validation.
7. The study emphasizes diagnosis and prognosis but neglects to explore potential correlations between methylation patterns and therapeutic responses, such as those to hypomethylating drugs. Emphasizing such correlations would enhance its clinical significance.
 8. The involvement of pediatric cohorts necessitates the examination of ethical issues associated with machine learning-based stratification in vulnerable populations.
 9. The scalability and cost challenges of long-read sequencing, especially in resource-constrained environments, are inadequately addressed in the study, despite its efficacy as a tool. Examining these obstacles and their remedies would yield a comprehensive viewpoint.
 10. The study incorporates data from several platforms; nevertheless, the lack of comprehensive methodologies for addressing batch effects or platform-specific biases undermines confidence in data harmonization.

Minor concerns:

1. The functionality of the web-based ALMA tool for non-computational doctors requires clarification. Elements such as user training or streamlined interfaces may be essential to facilitate adoption.
 2. Figures and Supplementary Materials. Certain figures, especially those depicting machine learning operations, are deficient in comprehensive captions and annotations. This may impede comprehension for a wide audience.
 3. The clarifications of terminology. Introducing essential computational terminology such as PaCMAP and LightGBM sooner in the paper and succinctly elucidating their significance would enhance accessibility.
 4. Supplementary Dataset Details. The publication should offer a more comprehensive analysis of the cohorts utilized, encompassing demographic distributions, quality control protocols, and constraints in data harmonization.
 5. Potential for Multi-Omics Integration. While outside the purview of this work, a brief discussion on the integration of methylation profiling with transcriptome or proteomic data could offer avenues for future research.
- In conclusion, this publication signifies a significant advancement in AML diagnosis and prognostics through the integration of sophisticated sequencing technology, machine learning, and extensive epigenomic data. The discoveries can transform leukemia management. Nevertheless, other elements necessitate additional scrutiny, such as prospective validation, model interpretability, and deliberations on the problems of clinical implementation. Addressing these issues would markedly improve the manuscript's influence and applicability.

(Remarks on code availability)

Version 1:

Reviewer comments:

Reviewer #1

(Remarks to the Author)

The authors comprehensively addressed the reviewer's criticism of their manuscript on epigenomic long-term diagnosis and prognosis in AML. They have clarified the handling of peripheral blood and bone marrow samples and demonstrated a high concordance of methylation signatures despite different blast counts. The validation cohort has been expanded through external collaboration and the limitations in classifying rare subtypes are now explicitly recognized. Issues with referencing of figures and cohort diversity have been resolved, with demographic representation and sampling biases transparently presented. Inconsistencies in performance metrics have been explained and more appropriate metrics provided. Prognostic models were tested and found to be robust in both pediatric and adult AML cohorts. The relationship between epigenomic and genetic risk models, including MRD1 status, was clarified, demonstrating the independence and added value of the epigenomic models.

The rationale for the 38CpG signature was explained and the arbitrary nature of the p-value threshold was recognized, with two different prediction models benchmarked. Alternative machine learning algorithms were compared and LightGBM was selected for its superior performance, although future deep learning approaches are planned. The biological significance of the CpGs was described in detail and statistical comparisons were made with established prognostic indicators, confirming the independent prognostic value of the epigenomic models. Details on nanopore sequencing, including depth and quality, were added, demonstrating high data quality and cross-platform concordance. The limitations of imputation and small nanopore sample size were acknowledged, and plans for methodological improvements and larger sample sizes were provided.

Overall, the authors have thoroughly addressed all major and minor concerns, supplemented the manuscript with additional data and analyses, and transparently discussed the remaining limitations. The manuscript is now suitable for acceptance.

(Remarks on code availability)

Reviewer #3

(Remarks to the Author)

The authors have successfully addressed all my comments.

(Remarks on code availability)

REVIEWER COMMENTS

Reviewer #1, expertise in DNA methylation and prognostic biomarkers for AML (Remarks to the Author):

The manuscript, "Long-read epigenomic diagnosis and prognosis of Acute Myeloid Leukemia," introduces the Acute Leukemia Methylome Atlas (ALMA), a diagnostic model integrating harmonized DNA methylation data from 3,314 patients. ALMA predicts 27 WHO 2022 acute leukemia subtypes with high accuracy (96.3% in discovery and 90.1% in validation cohorts). Two prognostic tools were developed: the AML Epigenomic Risk model, leveraging whole epigenome data for survival prediction, and the 38CpG AML Signature, a concise classifier based on 38 CpG sites.

The study also presents a rapid, long-read nanopore sequencing workflow enabling simultaneous genomic and epigenomic analysis, achieving accurate diagnosis and prognosis predictions within 24 hours. These findings highlight the clinical potential of epigenomic profiling for improved AML subtype stratification and personalized therapeutic approaches.

This study is a comprehensive and generally high-quality study. Nevertheless, there are some points of criticism and concerns that should be addressed by the authors.

Author Response: We are grateful for reviewer 1's thoughtful and insightful comments. We believe the edits suggested by the reviewer made the work substantially stronger and hope we have addressed them all accurately and completely.

- The study does not clearly address the differences between peripheral blood (PB) and bone marrow (BM) samples and how variations in blast count may impact the quantitative DNA methylation values and the resulting signatures.

Author Response: We acknowledge the importance of addressing differences in the specimen type. Unfortunately, though all the methylation data was from specimen obtained at diagnosis, the source information for the publicly available training dataset was not available. The manuscript now explicitly discusses results from paired PB/BM samples for the samples (page 12, lines 301-310).

In addition, Supplementary Table 6 highlights result in PB/BM with various blast percentages. We demonstrate high concordance in epigenomic classification despite differences in blast counts.

- The validation cohort for the ALMA 5-dimensional model is relatively small, which may undermine the robustness and generalizability of the findings.

Author Response: To address this, we have created a python package (<https://github.com/f-marchi/ALMA-classifier>), shared it with collaborators from Sweden, who independently tested our models in more >100 new patients from their NOPHO AML trials and returned the results. All results have been added to the paper. Additionally, within the timeframe of the revision process we increased our specimen-to-result sample number from 12 to 20.

- ALMA shows promise in stratifying subtypes, but the authors do not adequately address potential challenges in applying it to rare subtypes or ambiguous cases.

Author Response: We agree with the reviewer that there are some limitations in classifying rare subtypes (e.g., FUS::ERG fusion and Down Syndrome AML). We have now explicitly acknowledged them in results (page 12, lines 393-300) and discussion (page 14, lines 352-362). The revised manuscript emphasizes the need for expanded methylation datasets on these populations and continuous model updating as more data becomes available.

- Figure referencing appears to be incorrect, particularly for WHO discovery/validation data in Figures 2A/C and unknown discovery/validation data in Figures 2B/D.

Author Response: We agree that our Fig. 2 layout was confusing. We have refactored Fig. 2 and corrected all referencing errors. Thank you!

- While the study uses a large dataset from multiple cohorts, it is unclear how well it captures the diversity of AML subtypes and patient demographics. Potential sampling biases, such as geographic, age, or ethnic representation, should be considered.

Author Response: The revised manuscript provides clearer context regarding cohort composition, explicitly addressing potential geographic, age-related, and ethnic biases (now in Supplementary Table 2). Unfortunately, we can only use the data that has been collected and made publicly available. These data are predominantly from European ethnicity. However, we have sought to exclude CpGs known to correlate with ancestry (PMID: 27924034), and added a tab called "Race or ethnic group" to our interactive map (<https://f-marchi.github.io/UF-LambaLab-ALMA-app/>) to allow users transparent understanding of how each race is represented among the clusters that form ALMA.

- Metrics such as macro and weighted F1 scores show significant differences between discovery and validation cohorts. These discrepancies are not sufficiently explained.

Author Response: The macro F1 caused confusion because it was trying to evaluate subtypes with sample size of 0. This was not an appropriate metric, so we thank the reviewer for catching it. Figure 2 now shows a more granular and rigorous table of discovery, AML02,08 test, and (new) NOPHO AML test metrics.

- The potential for overfitting is not discussed, despite the high accuracy metrics observed in the discovery phase.

Author Response: The negligible differences in predictive performance metrics and statistical guarantees between training and independent test cohorts provides evidence of minimal overfitting (Fig. 2a). This evidence is supported by the new NOPHO AML trial testing we added in the revised version.

- The prognostic model is evaluated only in the pediatric AML sub-cohort. The authors should explain whether the findings hold true for adult AML populations.

Author Response: We thank the reviewer for this comment. To address this important point, prognostic models were tested on publicly available adult AML populations (TCGA, Beat AML cohorts). These results now included in the updated manuscript and confirm consistent predictive performance in adults and pediatric populations. They can be found in the newly added Supplementary Figs 3 and 5, also shown below.

Supplementary Fig. 3: OS outcomes by AML Epigenomic Risk in NOPHO AML, TCGA, and Beat AML cohorts. a NOPHO AML (n=111). **b** TCGA (n=194). **c** Beat AML Consortium (n=198).

Supplementary Fig. 5: OS outcomes by 38-CpG AML Signature in NOPHO AML, TCGA, and Beat AML cohorts.

a NOPHO AML (n=111). **b** TCGA (n=194). **c** Beat AML Consortium (n=198).

- The association between achieving MRD1 and genetic risk is not explored. The comparison between epigenomic and genetic models should be clarified, particularly in how risk groups are defined.

Author Response: We have analyzed achieving MRD1 and score, As shown below patients with high score show greater MRD1 positivity as compared to those in the low score group both in the discovery (High vs. Low MRD1 positive 38% vs. 25%, $p < 0.001$) and validation cohort (49% vs 33%, $p=0.03$) demonstrating the impact of epigenomics score on MRD1 this information has been added in the manuscript page 5, line147). Our manuscript also includes detailed forest plots comparing our prognostic models to MRD1 status. We established that epigenomic risk models remained independent predictors of outcomes even after adjusting for MRD1, FLT3, leukocyte counts, and other standard genetic-risk factors (Fig. 4 and 7). Besides these models have also included additional analysis results shown in Fig. 5 for AML Epigenomic risk groups and Supplementary Fig. 6 for 38cpG signature where Kaplan-Meier plots for OS show, that epigenomic factors continue to play a significant role, as there these are able to further classify patients with high- and low-risk groups even in the presence or absence of a positive MRD1 status.

a) Discovery cohort

b) AML02, 08 Test cohort

Fig. 5: Patient outcomes by Epigenomic Risk groups within MRD1 positive and MRD1 negative groups

Box plots showing abundance of MRD1 positive and negative patients within *AML Epigenomic Risk* groups. OS by MRD1 positive and MRD1 negative groups and OS within MRD1 positive and within MRD1 negative groups by *AML Epigenomic Risk* groups in **a)** discovery cohort and **b)** AML02,08 test cohort. MRD 1, minimal residual disease at end of first induction.

a Discovery cohort

b AML02, 08 Test cohort

Supplementary Fig 6: Patient outcomes by 38-CpG AML Signature groups within MRD1 positive and MRD1 negative groups

OS within MRD1 positive and within MRD1 negative groups by **38-CpG AML Signature groups in a)** discovery cohort and **b)** AML02,08 test cohort.

- For the 38CpG AML signature, the authors do not discuss whether adding or removing CpGs could replicate the performance of classical risk models.

Author Response: The manuscript discusses the rigorous process that was taken to filter the top 200 CpGs using a risk-adjusted, CoxPH-EWAS. These 200 were subsequently filtered to the final 38-CpG signature using a cross validated penalized term to the loss function (L1 regularization or LASSO). We believe it would not have been best scientific rigor to deliberately add or remove CpGs after establishing our hypothesis.

- The choice of CpGs for the 38CpG AML signature was based on a p-value threshold of $10e-5$, which could be arbitrary and exclude biologically relevant CpGs.

Author Response: The p-value threshold ($10e-5$) was selected in advance to balance statistical rigor and biological relevance. However, we fully agree that $10e-5$, or any specific p-value threshold, is arbitrary and could exclude biologically relevant CpGs. For this and other reasons, we describe and benchmark two prognostic models that are distinct in their development but reach similar performance.

- While the LightGBM model is efficient, it is unclear whether the authors compared its performance to other methods, such as deep learning, which might better capture complex patterns.

Author Response: Comparative analyses of alternative algorithms (Random Forest, XGBoost, SVM, Logistic Regression, and other models available through sklearn API) were performed, including a basic multi-layer perceptron. LightGBM demonstrated superior interpretability, accuracy, and computational efficiency, which prompted us to use it for the present work. However, we fully agree with the reviewer that an appropriate deep learning architecture may overpower LGBM. It was out of the scope of this work, but future versions of ALMA may be based on a multi-stage, self-supervised transformer model augmented with a masked autoencoding strategy. We were not able to pack such a complex algorithm in this paper, but we will be developing it open-source upon publication.

- The link between the CpGs in the 38CpG AML signature and underlying genes or biological mechanisms is not sufficiently addressed.

Author Response: A section detailing the biological significance of CpGs in the 38-CpG signature has been added, explicitly discussing their roles in leukemia tumorigenesis and fate. (page 16, lines 388-402). Additionally, we can provide a correlation matrix to the reviewer of the 38 CpGs and correlated genes as well as Pearson correlations for every depicted gene showing that there is significant correlation in many genes-CpG pairs, for example **RCBTB1 - cg17099306**: moderate negative linear correlation (Pearson correlation coefficient ($\rho = -0.4017$, $P = 0.0000$), **LINC01341-cg00532502**: moderate negative correlation ($\rho = -0.5289$, $P = 0.0001$), **KIF28P-cg00532502**: moderate negative correlation ($\rho = -0.6223$, $P = 0.0000$).

Correlation matrix of 38 CpGs and their genes, n=904

- The authors do not indicate whether statistical significance testing was performed to compare their proposed models with existing standards. The models predict survival at 5 years, but the outcomes are not compared with other established prognostic factors, such as FLT3 and NPM1 mutations. Integrating these molecular features could enhance predictive accuracy.

Author Response: The manuscript now clearly includes additional metrics such as sensitivity, specificity, predictive value, and ROC analyses (new Fig. 8). Direct comparisons with standard prognostic indicators (FLT3-ITD, AAML1831 risk groups) demonstrate how the epigenomic models enhance current prognostic frameworks. We established that epigenomic risk models remained independent predictors of outcomes even after adjusting for MRD1, FLT3, leukocyte counts, and other standard genetic-risk factors (Fig. 4 and Fig. 7).

We further established supplementary data for the reviewer indicating that the epigenomic risk model retains significant prognostic value within molecularly defined AML subgroups, specifically FLT3-ITD-positive and NPM1-positive AML. We do not have NPM1 data available in test cohort.

Low AML Epigenomic Risk, n=77					
At risk	77	59	52	32	0
Censored	0	2	4	11	29
Events	0	4	14	16	17
High AML Epigenomic Risk, n=88					
At risk	88	65	43	36	19
Censored	0	1	2	3	18
Events	0	22	43	49	51

Low 38CpG-AMLSignature, n=68					
At risk	68	60	52	45	27
Censored	0	2	4	9	26
Events	0	6	12	14	15
High 38CpG-AMLSignature, n=97					
At risk	97	76	50	43	24
Censored	0	1	2	5	21
Events	0	20	45	49	52

Low AML Epigenomic Risk, n=96					
At risk	96	87	83	78	0
Censored	0	0	2	8	86
Events	0	5	7	9	10
High AML Epigenomic Risk, n=71					
At risk	71	56	44	37	33
Censored	0	0	0	2	5
Events	0	15	27	32	33

Low 38CpG-AMLSignature, n=77					
At risk	77	72	69	64	59
Censored	0	0	2	6	11
Events	0	5	6	7	7
High 38CpG-AMLSignature, n=90					
At risk	90	75	62	56	52
Censored	0	0	0	0	2
Events	0	15	28	34	36

Low AML Epigenomic Risk, n=55					
At risk	55	48	44	41	24
Censored	0	3	5	8	25
Events	0	4	6	6	6
High AML Epigenomic Risk, n=36					
At risk	36	27	23	22	15
Censored	0	0	1	2	8
Events	0	9	12	12	13

Low 38CpG-AMLSignature, n=36					
At risk	36	31	30	29	18
Censored	0	2	3	4	15
Events	0	3	3	3	4
High 38CpG-AMLSignature, n=55					
At risk	55	44	37	34	21
Censored	0	1	3	6	18
Events	0	10	15	15	16

- The manuscript does not provide sufficient details about the sequencing coverage/depth or genetic findings derived from Nanopore sequencing.

Author Response: Nanopore sequencing depth and coverage details are now comprehensively added in Supplementary Table 6. Genetic findings, including structural variant detection, mutations, and fusions, are now explicitly detailed. Additionally, Fig. 9 shows IGV plots on the fusions and mutations of selected patents specimens evaluated using 38 nanopore.

- The authors do not evaluate whether Nanopore sequencing might introduce errors in methylation calling due to technical limitations.

Author Response: While the potential for error in methylation calling due to technical limitations of Nanopore sequencing exists, Supplementary Table 6, mean read quality column, shows that the quality of the sequencing data is sufficiently high to mitigate these concerns with mean read quality consistently ranging between 21 and 27, indicating that the data generated was of adequate quality for accurate methylation detection. Additionally, we can provide comparisons to array-based results, with demonstrated cross-platform concordance in one sample (AML02_150), which was tested both as array and nanopore and yielded identical prediction results. (p.10, lines 253) Our belief is that DNA methylation, unlike RNA or proteins, has the advantage for being a simple fraction measurement, which leads to very reproducible results even in distinct platform chemistries.

- Imputation of missing values using the mean CpG values could introduce bias, particularly for rare subtypes or outliers.

Author Response: We agree with the reviewer that this is an important limitation. We are releasing alma-classifier in v0.1.3, but in future versions we will remove the imputation step and use algorithms that natively accept missing values both in training and inference (we are thinking of a masked autoencoding strategy). This will be important because to the reviewer's point, genomic deletions may hold critical biological roles.

- The Nanopore analysis is based on very few samples. The cohort size should be increased, and patient follow-up data should be included to evaluate prognostic accuracy.

Author Response: We have now increased the nanopore sample size to 20 samples (from 12) and added substantial new data supporting them on Supplementary Table 6, results, and discussion. Many of these samples were provided prospectively as patients presented to the clinic, which does not give us the option of 5-year outcome data.

- Claims of a "close match" between model predictions and pathology are questionable. For example, AML with MDS changes maps to "otherwise normal control," and biphenotypic AML maps to "MDS-related."

Author Response: The revised manuscript clarifies misclassifications, explaining rare subtype misclassification and providing a transparent discussion of limitations. (Results: p.12-13, lines 304-326, discussion: p.15, lines: 374-384) Overall high concordance for common subtypes remains substantiated by genomic confirmations described in Fig. 9 and Supplementary Table 6.

- The language of the manuscript is generally strong and concise; however, minor adjustments are needed:

Author Response: We thank reviewer for the comment and have updated the manuscript as per recommendations.

- The title is somewhat misleading, as Nanopore longread sequencing makes up a rather small part of the manuscript.

Author Response: We agree and have removed "long-read" from the title.

- The manuscript does not adequately discuss limitations, including:
 - o The small cohort size for rare subtypes.
 - o Potential biases in methylation array data.
 - o Prognostic data limited to the pediatric cohort.

Author Response: To address these, we have added a new test cohort including >100 new patient samples and tested our models in adult cohorts as well. Limitations in classifying rare subtypes (e.g., FUS::ERG fusion and Down Syndrome AML) are now explicitly acknowledged in results (p. 11-12, lines 279-300), discussion (p.15, lines 352-362).

Reviewer #3, expertise in machine learning and prediction models (Remarks to the Author):

Author Response: We are grateful for reviewer 3's thoughtful and insightful comments. We believe the edits suggested by the reviewer made the work substantially stronger and hope we have addressed them all accurately and completely.

Evaluation Report for Manuscript "Long-read Epigenomic Diagnosis and Prognosis of Acute Myeloid Leukemia"

Summary of the Manuscript:

This publication examines significant obstacles in the diagnosis and prognosis of Acute Myeloid Leukemia (AML), a condition for which existing clinical techniques frequently lack precision and scalability. The authors propose a paradigm shift by integrating long-read sequencing technologies with DNA methylation profiling to address limitations in current methodologies, including incomplete diagnostic coverage, significant variability across datasets, and restricted interpretability of machine learning models. Although the study is unique and methodologically sound, it is crucial to identify aspects that necessitate further elucidation or enhancement to optimize its clinical and translational significance.

The article introduces a novel method to enhance the diagnostic and prognostic framework of Acute Myeloid Leukemia (AML) by utilization of long-read sequencing technology and DNA methylation profiling. The Acute Leukemia Methylome Atlas (ALMA) is a comprehensive DNA methylation database that accurately distinguishes AML subtypes, achieving 96.3% in the discovery cohort and 90.1% in the validation cohort. Innovative prognostic models, such as AML Epigenomic Risk and a 38-CpG signature, demonstrate robust predictive accuracy for overall survival (OS) and event-free survival (EFS). Comprehensive whole-genome and epigenome sequencing were amalgamated utilizing long-read technologies and validated across 3,314 patients from 11 heterogeneous cohorts. The study emphasizes the value of DNA methylation profiling in improving AML diagnoses and prognostics, providing a more accurate and scalable framework for precision oncology. The open-source web-based tool associated with ALMA also supports clinical and scientific applications, hence emphasizing the study's practical significance.

Major Concerns:

1. Although the models were verified using retrospective datasets, their efficacy in prospective, real-world multicenter studies has yet to be evaluated. This stage is essential for facilitating clinical adoption. The study fails to address the representativeness of the datasets about global AML populations, especially in low-resource or underrepresented contexts. Mitigating these constraints would strengthen the validity of the conclusions.

Author Response: We acknowledge the need for prospective, multicenter studies. The current analysis is critical towards increasing clinical appetite for a larger prospective study of this type. With results from multi-center AML16 soon to be released and AAML1831 been recently described, we are working to incorporate these techniques into future multi-center trial design. We also fully agree with the reviewer's point regarding low-resource or underrepresented contexts. This concern informed our choice of nanopore sequencing over other technologies, since it is the only sequencer at present that can cost as low as \$2,000 while providing comprehensive whole genome sequencing. We expect that in the next decade, the growing body of evidence will only continue to push clinicians worldwide towards performing WGS on all oncology patients. We used the cheapest technology that accomplishes that.

2. The implementation of long-read sequencing in standard diagnostics may encounter

obstacles related to cost, infrastructure, and technical knowledge. A cost-benefit analysis and a comparison with current diagnostic techniques would enhance the discourse. Regulatory procedures for sophisticated sequencing diagnostics must be evaluated, including potential barriers to approval or harmonization.

This is an important concern that the reviewer has raised. Current evidence suggests that we are entering a new era in diagnostic oncology (PMID: 34921008) that will substantially change how care is delivered and what tools are used. To this end, others have started to evaluate these questions at larger scales (PMID: 38956197). Though we believe that specific economic or infrastructure analyses were beyond the scope and focus of our present study, we have provided in the revised manuscript technical datapoints that will be critical to cost-benefit analyses: Nanopore sequencing depth and coverage details are now added in Supplementary Table 6. Genetic findings, including structural variant detection, mutations, and fusions, are now explicitly detailed. Additionally, new Fig. 6 shows IGV plots on the fusions and mutations of selected nanopore samples, suggesting, as proof-of-concept, that epigenomic diagnosis and prognosis may be successful at much lower coverages than necessary for genomic lesion identification, which means a decrease in cost per diagnosis.

3. Despite their efficacy, techniques such as PaCMAP and LightGBM frequently face criticism for being "black box" models. Incorporating a discourse on feature significance or interpretable results may reconcile the disparity between computational intricacy and therapeutic relevance.

Although PaCMAP and LightGBM are inherently complex, they are low-distortion techniques that are meant to preserve the structure of the high dimensional data at a lower dimensional space. Unlike a neural network, which is popular today, PaCMAP enables the visualization of a complex data through a simple map (<https://f-marchi.github.io/UF-LambaLab-ALMA-app/>). This map shows the knowledge boundaries, strengths and weaknesses of the dataset and algorithms used. Between low-efficacy but transparent techniques (logistic regressions) and high-efficacy black box techniques (neural networks), we aimed to be in the middle. We also created the 38 CpG signature to address this concern and offer as a fully transparent alternative (Supplementary Table 4). The revised manuscript includes a new, dedicated section detailing the biological significance of genes involved in the 38-CpG signature, explicitly discussing their roles in leukemia tumorigenesis and fate.

4. The diagnostic framework's elevated sensitivity may result in overdiagnosis or misclassification, especially in borderline instances. Implementing protections or rules would alleviate this risk for the document.

The reviewer's comment here prompted important improvements present in the revised manuscript: risks related to diagnostic sensitivity are now explicitly addressed, emphasizing the introduction of the following clinical decision safeguard:

- If confidence is below threshold (50%), the model now outputs "Not confident".
- For predictions with confidence between 50-80%, the model also outputs the second most likely subtype and its probability.

- All predictions can be checked by complementary genomic validation to minimize misclassification, as shown in newly added Fig. 9.

With these guardrails, we believe that low-confidence or wrong predictions can be promptly detected, which will be vital in uniquely complex karyotypes and edge cases (as shown in Supplementary Table 6).

5. Although hazard ratios (HRs) for survival are persuasive, supplementary performance measures, including sensitivity, specificity, and predictive value, must be presented to comprehensively evaluate the clinical relevance of the prognostic models. An elucidation of how the 38-CpG signature correlates with or enhances established prognostic indicators such as FLT3-ITD or NPM1 mutations would contextualize its significance.

The manuscript now clearly includes additional metrics such as sensitivity, specificity, predictive value, and ROC analyses (new Fig. 5). Direct comparisons with standard prognostic indicators (FLT3-ITD, AAML1831 risk groups) demonstrate how the epigenomic models enhance current prognostic frameworks. We established that epigenomic risk models remained independent predictors of outcomes even after adjusting for MRD1, FLT3, leukocyte counts, and other standard genetic-risk factors (Fig. 3 and Fig. 7).

6. Selection of Features for 38-CpG Signature. The article outlines the EWAS-Cox PH-LASSO pipeline but fails to elucidate the criteria for selecting the final set of CpGs and whether any bias may have affected this decision. The text lacks adequate information regarding the statistical criteria for significance, such as p-values for CpG inclusion. A clear justification for these thresholds and their biological importance is required. As well as for the data harmonization. Although the publication amalgamates data from 11 cohorts, it fails to address the mitigation of batch effects or discrepancies between platforms (e.g., Illumina EPIC versus 450K arrays). This is essential for guaranteeing dependable model training and validation.

Clear statistical criteria for selecting the 38-CpG signature, including rigorous cross-validation, risk-adjustment, regularization, and Bonferroni correction were clearly and empirically detailed in the first version (Fig. 6). Batch correction between cohorts was also performed according to PMID: 16632515 and detailed in Supplementary Fig. 10. Additionally, no discrepancies between Illumina EPIC vs. 450K arrays exist because these platforms share identical probe chemistries (PMID: 27924034), and only overlapping probes were used (452453 CpGs). Full description can be found in chapter 1 of the paper's electronic notebook https://f-marchi.github.io/ALMA/source/1_discovery

7. The study emphasizes diagnosis and prognosis but neglects to explore potential correlations between methylation patterns and therapeutic responses, such as those to hypomethylating drugs. Emphasizing such correlations would enhance its clinical significance.

This assessment is well taken since HMAs are commonly used in adult AML and expect to soon be approved as priming agents for pediatrics as well. Unfortunately, currently, there is no publicly available cohort of high-quality methylation patient samples before and after HMAs. The multi-center AML16 will eventually make those samples available, at which point this important analysis can be performed. We have, however, included a patient (uf1829) as a case study.

They presented with relapsed KMT2A-r AML while on Azacitidine. We show on the newly added Fig. 9 that both epigenomic and genomic lesions were identified successfully.

8. The involvement of pediatric cohorts necessitates the examination of ethical issues associated with machine learning-based stratification in vulnerable populations.

It is unclear to us what specifically is requested in this comment. Our work describes new technology meant to empower clinicians worldwide with advanced molecular diagnosis and prognosis. Our work is observational, and future efforts will be observational until unequivocal evidence supports otherwise.

9. The scalability and cost challenges of long-read sequencing, especially in resource-constrained environments, are inadequately addressed in the study, despite its efficacy as a tool. Examining these obstacles and their remedies would yield a comprehensive viewpoint.

We acknowledge this concern. As we describe in our response to comment number 2 and in the discussion of our revised manuscript, we utilized long-read sequencing because it is the only technology at present capable of detecting complex structural variants and duplications, both of which escape short read and PCR capabilities. These variant types are diagnostic for many AML subtypes, such as KMT2A-r, which may fuse with more than >100 different genes at different loci. Additionally, as whole genome sequencing inevitably moves to standard of care in oncology, this technology enables concomitant epigenome sequencing natively, which adds another omics layer of information without any extra cost. Between long-read sequencing technologies, we chose Oxford Nanopore because their MinION sequencer costs \$2000, which is orders of magnitude more affordable than PacBio's new Vega sequencer (\$170,000). It is also far more affordable than most microscopes, flow cytometers, and other short read sequencers. We agree, however, that clinical genomics as a field faces important scalability challenges. To contribute to the solution, we have revised our work to include technical sequencing details such as coverage, N50, and accuracy scores for each of the 20 patient samples processed (Supplementary Table 6. Fig. 9 also aims to show that confident diagnoses can be achieved at coverages as low as 1x, which offers substantial reduction in sequencing costs when compared to standard 30x WGS.

10. The study incorporates data from several platforms; nevertheless, the lack of comprehensive methodologies for addressing batch effects or platform-specific biases undermines confidence in data harmonization.

The revised manuscript acknowledges potential errors inherent in Nanopore methylation calling, citing literature and providing comparisons to array-based results, with demonstrated cross-platform concordance in one sample (AML02_150), which was tested both as array and nanopore and yielded identical prediction results. Our belief is that DNA methylation, unlike RNA or proteins, has the advantage for being a simple fraction measurement, which leads to very reproducible results even in distinct platform chemistries. Also, as we describe in our response to comment number 6: batch correction between cohorts was performed according to PMID: 16632515 and detailed in Supplementary Fig. 10. Additionally, no discrepancies between Illumina EPIC vs. 450K arrays exist because these platforms share identical probe chemistries (PMID: 27924034), and only overlapping probes were used (452453 CpGs). Full description can

be found in chapter 1 of the paper's electronic notebook https://f-marchi.github.io/ALMA/source/1_discovery

Minor concerns:

1. The functionality of the web-based ALMA tool for non-computational doctors requires clarification. Elements such as user training or streamlined interfaces may be essential to facilitate adoption.

We recognize the importance of usability for clinical practitioners. The ALMA web-app provides interactive visualization with intuitive features like zooming, selecting, dragging, and downloading data subsets. In future versions, we will add brief user instructions and examples of clinical workflows to clarify usage, facilitating adoption among non-computational clinicians.

2. Figures and Supplementary Materials. Certain figures, especially those depicting machine learning operations, are deficient in comprehensive captions and annotations. This may impede comprehension for a wide audience.

We agree that the clarity of machine learning figures in the first submission was lacking. We thank the reviewer for the suggestion and have completely refactored the figures (especially Fig. 2) to improve clarity and transparency.

3. The clarifications of terminology. Introducing essential computational terminology such as PaCMAP and LightGBM sooner in the paper and succinctly elucidating their significance would enhance accessibility.

We appreciate this suggestion. Essential terms such as PaCMAP and LightGBM are now introduced appropriately along with the changes to match the formatting requirements of the journal.

4. Supplementary Dataset Details. The publication should offer a more comprehensive analysis of the cohorts utilized, encompassing demographic distributions, quality control protocols, and constraints in data harmonization.

In the revised manuscript, we have added a new Supplementary Table 2 containing comprehensive patient characteristics that include detailed demographics, as requested.

5. Potential for Multi-Omics Integration. While outside the purview of this work, a brief discussion on the integration of methylation profiling with transcriptome or proteomic data could offer avenues for future research.

We agree on the significance of multi-omics integration. As discussed, we believe that the most feasible way to add new omic layers to clinical workflows is if they come as a free combo. Our sequencing pipeline provides two omics concomitantly: genomics and epigenomics. The revised manuscript includes extensive new results and discussion on the genomic data generated. Additionally, Supplementary Fig. 8 combines our models with risk group and previously published transcriptomic prognostic signatures, suggesting promising complementary value.

Once again, we thank the reviewer for the very thoughtful and constructive comments.

In conclusion, this publication signifies a significant advancement in AML diagnosis and prognostics through the integration of sophisticated sequencing technology, machine learning, and extensive epigenomic data. The discoveries can transform leukemia management. Nevertheless, other elements necessitate additional scrutiny, such as prospective validation, model interpretability, and deliberations on the problems of clinical implementation. Addressing these issues would markedly improve the manuscript's influence and applicability.